# Variational Bayesian Last Layers

**James Harrison[1], John Willes[2], Jasper Snoek[1]**
[1]Google DeepMind, [2]Vector Institute
`jamesharrison@google.com, john.willes@vectorinstitute.ai,`
`jsnoek@google.com`

## Abstract

We introduce a deterministic variational formulation for training Bayesian last layer neural networks. This yields a sampling-free, single-pass model and loss that effectively improves uncertainty estimation. Our variational Bayesian last layer (VBLL) can be trained and evaluated with only quadratic complexity in last layer width, and is thus (nearly) computationally free to add to standard architectures. We experimentally investigate VBLLs, and show that they improve predictive accuracy, calibration, and out of distribution detection over baselines across both regression and classification. Finally, we investigate combining VBLL layers with variational Bayesian feature learning, yielding a lower variance collapsed variational inference method for Bayesian neural networks.

## 1 Introduction

Well-calibrated uncertainty quantification is essential for reliable decision-making with machine learning systems. However, many methods for improving uncertainty quantification in deep learning (including Bayesian methods) have seen limited application due to their relative complexity over standard deep learning. For example, methods such as sampling-based mean field variational inference (Blundell et al., 2015), Markov chain Monte Carlo (MCMC) methods (Papamarkou et al., 2022; Neal, 1995; Izmailov et al., 2021), and comparatively simple heuristics such as Bayesian dropout (Gal & Ghahramani, 2016) all have substantially higher computational cost than baseline networks. Single-pass methods (where only one network evaluation is required) often require substantial modifications to network architectures, regularization, or training and evaluation procedures, even for the simplest such models (Liu et al., 2022; Wilson et al., 2016b; Kristiadi et al., 2021).

In this work, we take a simplicity-first approach to Bayesian deep learning, and develop a conceptually simple and computationally inexpensive partially Bayesian neural network. In particular, we investigate variational learning of Bayesian last layer (BLL) neural networks. While BLL models consider only the uncertainty over the output layer of the network, they have been shown to perform comparably to more complex Bayesian models (Watson et al., 2021; Harrison et al., 2018; Fiedler & Lucia, 2023; Kristiadi et al., 2020). Our variational formulation relies on a deterministic lower bound on the marginal likelihood, which enables highly-efficient mini-batch, sampling-free loss computation, and is thus highly scalable.

**Contributions.** Concretely, the contributions of this work are:

- We present variational Bayesian last layers (VBLLs), a novel last layer neural network component for uncertainty quantification which can be straightforwardly included in standard architectures and training pipelines (including fine-tuning), for both deterministic and Bayesian neural networks.
- We derive principled and sampling-free Bayesian training objectives for VBLLs, and show that with careful parameterization they can be computed at the same cost as standard training, and trained with standard mini-batch training.
- We show that VBLLs improve predictive accuracy, likelihoods, calibration, and out of distribution detection across a wide variety of problem settings. We also show VBLLs strongly outperform baseline models in contextual bandits.
- We release an easy-to-use package providing efficient VBLL implementations in PyTorch.

## 2 Bayesian Last Layer Neural Networks

We first review Bayesian last layer models which maintain a posterior distribution only for the last layer in a neural network. These models correspond to Bayesian (linear or logistic) regression or Bayesian Gaussian discriminant analysis (for each of the three models we present, respectively) with

learned features. We assume $T$ total data points, and write inputs as $\boldsymbol{x} \in \mathbb{R}^{N_x}$. For regression, outputs are $\boldsymbol{y} \in \mathbb{R}^{N_y}$; for classification, outputs are $y \in \{1, \ldots, N_y\}$, and $\boldsymbol{y}$ denotes the $N_y$-dimensional one-hot representation. For all models discussed in this section, we will use neural network features $\boldsymbol{\phi} : \mathbb{R}^{N_x} \times \Theta \to \mathbb{R}^{N_\phi}$. These correspond to all parts of a network architecture but the last layer, where $\boldsymbol{\theta} \in \Theta$ denotes the weights of the neural network. We will typically write $\boldsymbol{\phi} := \boldsymbol{\phi}(\boldsymbol{x}, \boldsymbol{\theta})$ for notational convenience and refer to these parameters as *features* because they define the map from inputs to the feature embedding on which the BLL operates.

## 2.1 REGRESSION

The canonical BLL model for the regression case[1] is

$$\boldsymbol{y} = \boldsymbol{w}^\top \boldsymbol{\phi}(\boldsymbol{x}, \boldsymbol{\theta}) + \boldsymbol{\varepsilon} \tag{1}$$

where $\boldsymbol{\varepsilon}$ is assumed to be normally distributed with zero mean and covariance $\Sigma$, and these noise terms are i.i.d. across realizations. We specify a Gaussian prior[2] $p(\boldsymbol{w}) = \mathcal{N}(\underline{\boldsymbol{w}}, \underline{S})$, assumed independent of the noise $\boldsymbol{\varepsilon}$. Posterior inference in the BLL model is analytically tractable for a fixed set of features. The marginal likelihood may be computed either via direct computation or by iterating over the dataset. Fixing a distribution over $\boldsymbol{w}$ of the form $\mathcal{N}(\bar{\boldsymbol{w}}, S)$, the predictive distribution is

$$p(\boldsymbol{y} \mid \boldsymbol{x}, \boldsymbol{\eta}, \boldsymbol{\theta}) = \mathcal{N}(\bar{\boldsymbol{w}}^\top \boldsymbol{\phi}, \boldsymbol{\phi}^\top S \boldsymbol{\phi} + \Sigma) \tag{2}$$

where $\boldsymbol{\eta}$ denotes the parameters of the distribution, here $\boldsymbol{\eta} = (\bar{\boldsymbol{w}}, S)$.

## 2.2 DISCRIMINATIVE CLASSIFICATION

In this subsection we introduce a BLL model that corresponds to standard classification neural networks, where

$$p(\boldsymbol{y} \mid \boldsymbol{x}, W, \boldsymbol{\theta}) = \mathrm{softmax}(\boldsymbol{z}), \qquad \boldsymbol{z} = W \boldsymbol{\phi}(\boldsymbol{x}, \boldsymbol{\theta}) + \boldsymbol{\varepsilon} \tag{3}$$

where $\boldsymbol{z} \in \mathbb{R}^{N_y}$ are the logits. These are also interpreted as unnormalized joint data-label log likelihoods (Grathwohl et al., 2020), where

$$\boldsymbol{z} = \log p(\boldsymbol{x}, \boldsymbol{y} \mid W, \boldsymbol{\theta}) - Z(W, \boldsymbol{\theta}) \tag{4}$$

where $Z(W, \boldsymbol{\theta})$ is a normalizing constant, independent of the data. The term $\boldsymbol{\varepsilon} \in \mathbb{R}^{N_y}$ is a zero-mean Gaussian noise term with variance $\Sigma$. Typically in logistic regression this noise term is ignored, although it has seen use to model label noise (Collier et al., 2021). We include it to unify the presentation, and the variance can be assumed zero as necessary.

As in the regression case, we specify a Gaussian prior for $W$. In contrast with the regression setting, exact inference and computation of the posterior predictive is not analytically tractable in this model. We refer to this model—consisting of multinominal Bayesian logistic regression on learned neural network features—as discriminative classification, as logistic regression is a classical discriminative learning algorithm.

## 2.3 GENERATIVE CLASSIFICATION

The second classification model we consider is the *generative classification* model (Harrison et al., 2020; Zhang et al., 2021; Willes et al., 2022), so-called due to its similarity to classical generative models such as Gaussian discriminant analysis. In this model, we assume that the features associated with each class are normally distributed. Placing a Normal prior on the means of these feature distributions and a (conjugate) Dirichlet prior on class probabilities, we have priors and likelihoods (top line and bottom line respectively) of the form

$$\boldsymbol{\rho} \sim \mathrm{Dir}(\boldsymbol{\alpha}) \qquad\qquad \boldsymbol{\mu_y} \sim \mathcal{N}(\underline{\bar{\boldsymbol{\mu}}_{\boldsymbol{y}}}, \underline{S}_{\boldsymbol{y}}) \tag{5}$$

$$\boldsymbol{y} \mid \boldsymbol{\rho} \sim \mathrm{Cat}(\boldsymbol{\rho}) \qquad\qquad \boldsymbol{\phi} \mid \boldsymbol{y} \sim \mathcal{N}(\boldsymbol{\mu_y}, \Sigma). \tag{6}$$

In this model, $\underline{\bar{\boldsymbol{\mu}}}_{\boldsymbol{y}} \in \mathbb{R}^{N_\phi}$ and $\underline{S}_{\boldsymbol{y}} \in \mathbb{R}^{N_\phi \times N_\phi}$ are the prior mean and covariance over $\boldsymbol{\mu_y} \in \mathbb{R}^{N_\phi}$, the mean embedding for each. The subscript here indexes the statistics for each class; we also write $\boldsymbol{\mu} := \{\boldsymbol{\mu}_1, \ldots, \boldsymbol{\mu}_{N_y}\}$ to terms for all $\boldsymbol{y}$. The terms $\boldsymbol{\rho} \in \mathcal{P}_{N_y}$ correspond to class probabilities, where $\mathcal{P}_{N_y}$ denotes the probability simplex embedded in $\mathbb{R}^{N_y}$. These class probabilities are in turn used in the categorical distribution over the class.

---

[1]We present scalar-output regression in the paper body and defer the multivariate output case to the appendix.
[2]Throughout the paper, we use overbars to denote mean parameters and underbars to denote prior parameters.

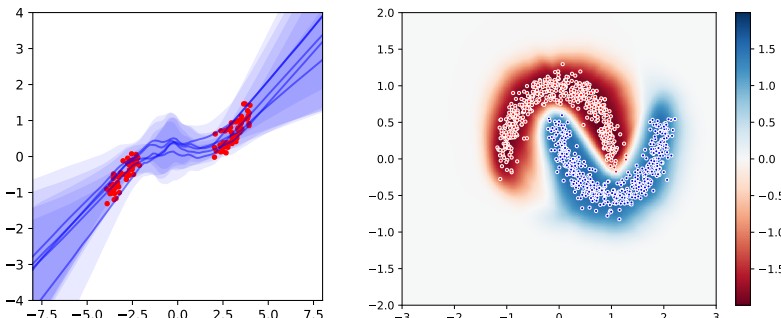

Figure 1: **Left**: A variational BLL (VBLL) regression model with BBB features trained on 50 data points generated from a cubic function with additive Gaussian noise. The plot shows the 95% predictive credible region under the variational posterior for several sampled feature weights. **Right**: Visualizing (re-scaled) $p(\boldsymbol{x} \mid \boldsymbol{y} = 1) - p(\boldsymbol{x} \mid \boldsymbol{y} = 0)$ predicted by a generative VBLL model on the half moon dataset, shows good sensitivity to Euclidean distance and sensible embedding densities.

For a distribution over model parameters

$$p(\boldsymbol{\rho}, \boldsymbol{\mu} \mid \boldsymbol{\eta}) = \text{Dir}(\boldsymbol{\alpha}) \prod_{k=1}^{N_y} \mathcal{N}(\bar{\boldsymbol{\mu}}_k, S_k) \tag{7}$$

for which we write $\boldsymbol{\eta} = \{\boldsymbol{\alpha}, \bar{\boldsymbol{\mu}}, S\}$, we have

$$p(\boldsymbol{x} \mid \boldsymbol{y}, \boldsymbol{\eta}) = \mathcal{N}(\bar{\boldsymbol{\mu}}_{\boldsymbol{y}}, \Sigma + S_{\boldsymbol{y}}), \qquad p(\boldsymbol{y} \mid \boldsymbol{\eta}) = \frac{\boldsymbol{\alpha}_{\boldsymbol{y}}}{\sum_{k=1}^{N_y} \boldsymbol{\alpha}_k} \tag{8}$$

via analytical marginalization. To compute the predictive over class labels, we apply Bayes' rule, yielding

$$p(\boldsymbol{y} \mid \boldsymbol{x}, \boldsymbol{\eta}) = \text{softmax}_{\boldsymbol{y}}(\log p(\boldsymbol{x} \mid \boldsymbol{y}, \boldsymbol{\eta}) + \log p(\boldsymbol{y} \mid \boldsymbol{\eta})). \tag{9}$$

Here,

$$\log p(\boldsymbol{x} \mid \boldsymbol{y}, \boldsymbol{\eta}) = -\frac{1}{2}((\boldsymbol{\phi} - \bar{\boldsymbol{\mu}}_{\boldsymbol{y}})^\top (\Sigma + S_{\boldsymbol{y}})^{-1} (\boldsymbol{\phi} - \bar{\boldsymbol{\mu}}_{\boldsymbol{y}}) + \text{logdet}(\Sigma + S_{\boldsymbol{y}}) + c) \tag{10}$$

where $c$ is a constant, shared for all classes, that may be ignored due to the shift-invariance of the softmax. Grouping the log determinant term with the class prior yields a bias term. Instead of a linear transformation of the input features to obtain a class logit, we instead have a quadratic transformation. This formulation is a strict generalization of standard classifier architectures (Harrison, 2021), in which we have quadratic decision regions as opposed to linear ones.

## 2.4 INFERENCE AND TRAINING IN BLL MODELS

BLL models have seen growing popularity in recent years, ironically driven in part by a need for compatibility with increasingly deep models (Snoek et al., 2015; Azizzadenesheli et al., 2018; Harrison et al., 2018; Weber et al., 2018; Riquelme et al., 2018; Harrison et al., 2020; Ober & Rasmussen, 2019; Kristiadi et al., 2020; Thakur et al., 2020; Watson et al., 2020; 2021; Daxberger et al., 2021a; Willes et al., 2022; Sharma et al., 2022; Schwöbel et al., 2022; Zhang et al., 2021; Moberg et al., 2019; Fiedler & Lucia, 2023). Exact marginalization enables computationally efficient treatment of uncertainty, as well as resulting in lower-variance training objectives compared to sampling-based Bayesian models. A common and principled objective for training BLL models is the (log) marginal likelihood (Harrison et al., 2018), via gradient descent on

$$T^{-1} \log p(Y \mid X, \boldsymbol{\theta}) \tag{11}$$

where $X, Y$ denote stacked data. We include a factor of $T^{-1}$ to enable better comparison with standard, non-Bayesian, training pipelines (typically based on average loss over mini-batches) and across dataset sizes. This training objective can be problematic, however: gradient computation requires computing the full marginal likelihood, and mini-batches do not yield unbiased gradient estimators as in standard training with an arbitrary loss function. Even mini-batch processing of the dataset—iterating between conditioning on mini-batches and prediction under the partial posterior—induces long computation graphs that make training at scale impossible. Moreover, due to the flexibility of neural network features, a full marginal likelihood training objective can result in substantial over-concentration of the approximate posterior (Thakur et al., 2020; Ober et al., 2021).

## 3 SAMPLING-FREE VARIATIONAL INFERENCE FOR BLL NETWORKS

To exploit exact marginalization while avoiding full marginal likelihood computation, we will turn to stochastic variational inference (Hoffman et al., 2013). In particular, we aim to jointly compute an approximate last layer posterior and optimize network weights by maximizing lower bounds on marginal likelihood. As such, we will avoid distributional assumptions made in the previous section. We write the (uncertain) last layer parameters as $\boldsymbol{\xi}$ and aim to find an approximate posterior $q(\boldsymbol{\xi} \mid \boldsymbol{\eta})$ parameterized by $\boldsymbol{\eta}$. Concretely, throughout this section we will develop bounds of the form

$$T^{-1} \log p(Y \mid X, \boldsymbol{\theta}) \geq \mathcal{L}(\boldsymbol{\theta}, \boldsymbol{\eta}, \Sigma) - T^{-1}\mathrm{KL}(q(\boldsymbol{\xi} \mid \boldsymbol{\eta})||p(\boldsymbol{\xi})) \tag{12}$$

where $\mathcal{L}$ is architecture dependent and developed in the remainder of this section. Thus, practically, the $T^{-1}$ factor weights regularization terms in our training objective. In this section, we index data with $t$ (via subscript), including $\boldsymbol{\phi}_t := \boldsymbol{\phi}(\boldsymbol{x}_t, \boldsymbol{\theta})$.

### 3.1 REGRESSION

We consider the log marginal likelihood $\log p(Y \mid X, \boldsymbol{\theta})$, with marginalized parameters $\boldsymbol{\xi} = \{\boldsymbol{w}\}$, and have the following lower bound.

**Theorem 1.** *Let $q(\boldsymbol{\xi} \mid \boldsymbol{\eta}) = \mathcal{N}(\bar{\boldsymbol{w}}, S)$ denote the variational posterior for the BLL model defined in Section 2.1. Then, (12) holds with*

$$\mathcal{L}(\boldsymbol{\theta}, \boldsymbol{\eta}, \Sigma) = \frac{1}{T} \sum_{t=1}^{T} \left( \log \mathcal{N}(\boldsymbol{y}_t \mid \bar{\boldsymbol{w}}^\top \boldsymbol{\phi}_t, \Sigma) - \frac{1}{2}\boldsymbol{\phi}_t^\top S \boldsymbol{\phi}_t \Sigma^{-1} \right). \tag{13}$$

The proof for this result and all others is available in Appendix F. When $q(\boldsymbol{\xi} \mid \boldsymbol{\eta}) = p(\boldsymbol{\xi} \mid Y, X)$ and distributional assumptions are satisfied, this lower bound is tight (this may be shown by direct substitution). This correspondence between the variational and true posterior for appropriately-chosen variational families is well known—see Knoblauch et al. (2019) for a thorough discussion. We note that a similar objective for regression models was developed in Watson et al. (2021).

### 3.2 DISCRIMINATIVE CLASSIFICATION

In the discriminative classification case, the parameters are $\boldsymbol{\xi} = \{W\}$. We will assume a diagonal covariance matrix $\Sigma$, and write $\sigma_i^2 := \Sigma_{ii}$. We will fix a variational posterior of the form $q(W \mid \boldsymbol{\eta}) = \prod_{k=1}^{N_y} q(\boldsymbol{w}_k \mid \boldsymbol{\eta}) = \prod_{k=1}^{N_y} q(\bar{\boldsymbol{w}}_k, S_k)$, where $\boldsymbol{w}_k$ denotes the $k$'th row of $W$. This factorization retains dense covariances for each class, but sacrifices cross-class covariances. While we only present this factorized variational posterior, a similar training objective may be derived with a fully dense variational posterior. Under the variational posterior, we have the following bound on the marginal likelihood.

**Theorem 2.** *Let $q(W \mid \boldsymbol{\eta}) = \prod_{k=1}^{N_y} \mathcal{N}(\bar{\boldsymbol{w}}_k, S_k)$ denote the variational posterior for the discriminative classification model defined in Section 2.2. Then, (12) holds with*

$$\mathcal{L}(\boldsymbol{\theta}, \boldsymbol{\eta}, \Sigma) = \frac{1}{T} \sum_{t=1}^{T} \left( \boldsymbol{y}_t^\top \bar{W} \boldsymbol{\phi}_t - \mathrm{LSE}_k \left[ \bar{\boldsymbol{w}}_k^\top \boldsymbol{\phi}_t + \frac{1}{2}(\boldsymbol{\phi}_t^\top S_k \boldsymbol{\phi}_t + \sigma_k^2) \right] \right) \tag{14}$$

Here, $\mathrm{LSE}_k(\cdot)$ denotes the log-sum-exp function, with the sum over $k$. In contrast to the regression case, this lower bound is a lower bound on the standard ELBO (due to two applications of Jensen's inequality) and the bound is not tight. We have reduced variance (which would be induced by sampling logit values before the softmax in standard SVI (Ovadia et al., 2019)) for bias due to this lower bound. Our proof leverages the same double application of Jensen's inequality used by Blei & Lafferty (2007). We note that tighter analytically tractable lower bounds exist for the logistic regression model (Depraetere & Vandebroek, 2017; Knowles & Minka, 2011), although for simplicity of the resulting algorithm we use the above lower bound.

### 3.3 GENERATIVE CLASSIFICATION

In the generative classification case, the parameters are $\boldsymbol{\xi} = \{\boldsymbol{\mu}, \boldsymbol{\rho}\}$. In this setting, the Dirichlet posterior over class probabilities $p(\boldsymbol{\rho} \mid Y)$ can be computed exactly with one pass over the data by simply counting class occurrences. We therefore only consider a variational posterior of the form $q(\boldsymbol{\xi} \mid \boldsymbol{\eta}, Y) = q(\boldsymbol{\mu} \mid \boldsymbol{\eta})$ for the class embeddings, where $q(\boldsymbol{\mu} \mid \boldsymbol{\eta}) = \prod_{k=1}^{N_y} \mathcal{N}(\bar{\boldsymbol{\mu}}_k, S_k)$. This yields the following lower bound.

**Theorem 3.** *Let* $q(\boldsymbol{\mu} \mid \boldsymbol{\eta}) = \prod_{k=1}^{N_y} \mathcal{N}(\bar{\boldsymbol{\mu}}_k, S_k)$ *denote the variational posterior over class embeddings for the generative classification model defined in Section 2.3. Let* $p(\boldsymbol{\rho} \mid Y) = Dir(\boldsymbol{\alpha})$ *denote the exact Dirichlet posterior over class probabilities, with* $\boldsymbol{\alpha}$ *denoting the Dirichlet posterior concentration parameters. Then,* (12) *holds with*

$$\mathcal{L}(\boldsymbol{\theta}, \boldsymbol{\eta}, \Sigma) = \frac{1}{T} \sum_{t=1}^{T} \left( \log \mathcal{N}(\boldsymbol{\phi}_t \mid \bar{\boldsymbol{\mu}}_{\boldsymbol{y}_t}, \Sigma) - \frac{1}{2} \text{tr}(\Sigma^{-1} S_{\boldsymbol{y}_t}) + \psi(\boldsymbol{\alpha}_{\boldsymbol{y}_t}) - \psi(\boldsymbol{\alpha}_*) + \log \boldsymbol{\alpha}_* \right. \quad (15)$$

$$\left. - \text{LSE}_k[\log \mathcal{N}(\boldsymbol{\phi}_t \mid \bar{\boldsymbol{\mu}}_k, \Sigma + S_k) + \log \boldsymbol{\alpha}_k]\right)$$

*where* $\psi(\cdot)$ *is the digamma function and where* $\boldsymbol{\alpha}_* = \sum_k \boldsymbol{\alpha}_k$.

Importantly, we note that $\psi(\boldsymbol{\alpha}_{\boldsymbol{y}_t}), \psi(\boldsymbol{\alpha}_*), \log \boldsymbol{\alpha}_*$ all vanish in gradient computation and may be ignored. The term $\log \boldsymbol{\alpha}_k$ is the LSE can not be ignored, however. This training objective is again a lower bound on the ELBO, and is not tight. The first Dirichlet term (in the upper line) vanishes in gradient computation, but the second term inside the log-sum-exp function does not. In the case that the posterior concentration parameters are equal for all classes (as in the case of a balanced dataset), the concentration parameter can be pulled out of the LSE($\cdot$) (due to the equivariance of log-sum-exp under shifts) and can be ignored.

### 3.4 TRAINING VBLL MODELS

We propose three methods to learn VBLL models.

**Full training.** First, we can jointly optimize the last layer variational posterior together with MAP estimation of the features, yielding combined training objective

$$\boldsymbol{\theta}^*, \boldsymbol{\eta}^*, \Sigma^* = \underset{\boldsymbol{\theta}, \boldsymbol{\eta}, \Sigma}{\arg\max} \left\{ \mathcal{L}(\boldsymbol{\theta}, \boldsymbol{\eta}, \Sigma) + T^{-1}(\log p(\boldsymbol{\theta}) + \log p(\Sigma) - \text{KL}(q(\boldsymbol{\xi} \mid \boldsymbol{\eta}) \| p(\boldsymbol{\xi}))) \right\}. \quad (16)$$

While one may expect this to result in substantial over-concentration for weak feature priors, in practice we observe that stochastic regularization due to mini-batch optimization prevents overconcentration. Throughout this work, we will place simple isotropic zero-mean Gaussian priors on feature weights (yielding weight decay regularization) and a canonical inverse-Wishart prior on $\Sigma$. For Gaussian priors (as developed throughout this section) the KL regularization term can be computed in closed form. The prior terms (and the KL penalty) introduce a set of new hyperparameters that may be difficult to select. In Appendix C, we discuss these hyperparameters and their interpretation, and provide a reformulation of hyperparameters that increases interpretability.

**Post-training.** As an alternative to jointly optimizing the variational last layer with the features, a two step procedure can be used. In this step, the feature weights $\boldsymbol{\theta}$ are trained by any arbitrary training procedure (e.g. standard neural network training) and the last layer (and $\Sigma$) are trained with frozen features. The training objective is identical to (16), although $\boldsymbol{\theta}^*$ is trained in the initial pre-training step and $\boldsymbol{\eta}^*, \Sigma^*$ are trained via (16).

**Feature uncertainty.** Lastly, we can combine last layer SVI with variational feature learning (Blundell et al., 2015), corresponding to approximate collapsed VI (Teh et al., 2006). This training strategy allows us to construct a variational posterior on the full marginal likelihood, via

$$\log p(Y \mid X) \geq \mathbb{E}_{q(\boldsymbol{\xi}, \boldsymbol{\theta}, \Sigma \mid \boldsymbol{\eta})}[\log(Y \mid X, \boldsymbol{\xi}, \boldsymbol{\theta}, \Sigma)] - \text{KL}(q(\boldsymbol{\xi}, \boldsymbol{\theta}, \Sigma \mid \boldsymbol{\eta}) \| p(\boldsymbol{\xi}, \boldsymbol{\theta}, \Sigma)). \quad (17)$$

Assuming the prior and variational posterior factorize across the features and last layer, we can partially collapse this expectation

$$\mathbb{E}_{q(\boldsymbol{\xi}, \boldsymbol{\theta}, \Sigma \mid \boldsymbol{\eta})}[\log(Y \mid X, \boldsymbol{\xi}, \boldsymbol{\theta}, \Sigma)] = \mathbb{E}_{q(\boldsymbol{\theta}, \Sigma \mid \boldsymbol{\eta})} \mathbb{E}_{q(\boldsymbol{\xi} \mid \boldsymbol{\eta})}[\log(Y \mid X, \boldsymbol{\xi}, \boldsymbol{\theta}, \Sigma)] \geq T \mathbb{E}_{q(\boldsymbol{\theta}, \Sigma \mid \boldsymbol{\eta})}[\mathcal{L}(\boldsymbol{\xi}, \boldsymbol{\eta}, \Sigma)]$$

$$(18)$$

and the KL penalty may be similarly decomposed into several terms that can be computed in closed form under straightforward distributional assumptions. In the above, we have included $\Sigma$ in the variational posterior, although practically we perform MAP estimation of this covariance under inverse-Wishart priors. Again in this setting, pre-training and post-training steps may be combined, but we do not investigate this case.

### 3.5 PREDICTION WITH VBLL MODELS

For prediction in VBLL models, we will predict under the variational posterior directly, approximating (for test input/label $(\boldsymbol{x}, \boldsymbol{y})$),

$$p(\boldsymbol{y} \mid \boldsymbol{x}, X, Y) \approx \mathbb{E}_{q(\boldsymbol{\xi} \mid \boldsymbol{\eta}^*)}[p(\boldsymbol{y} \mid \boldsymbol{x}, \boldsymbol{\xi}, \boldsymbol{\theta}^*, \Sigma^*)] \quad (19)$$

for the deterministic feature model. This expectation may be computed in closed form (for the regression and generative classification model) due to conjugacy, and can be computed via inexpensive last layer sampling in the discriminative classification model. In the variational feature model,

$$p(\boldsymbol{y} \mid \boldsymbol{x}, X, Y) \approx \mathbb{E}_{q(\boldsymbol{\theta} \mid \boldsymbol{\eta}^*)} \mathbb{E}_{q(\boldsymbol{\xi} \mid \boldsymbol{\eta}^*)}[p(\boldsymbol{y} \mid \boldsymbol{x}, \boldsymbol{\xi}, \boldsymbol{\theta}, \Sigma^*)] \quad (20)$$

where the inner expectation may be computed exactly and the outer expectation may be approximated via sampling. Further details of training, prediction, and out of distribution detection within all three VBLL models is provided in Appendix B.

For both training and prediction, under relatively weak assumptions on covariance matrices, computational complexity (for the classification models) is at most[3] $\mathcal{O}(N_y N_\phi^2)$, and can be reduced to $\mathcal{O}(N_y N_\phi)$ for diagonal covariances. This matches the complexity of standard network evaluation; for reasonable choices of covariance sparsity, the additional computational cost of VBLL models over standard networks is negligible. More details are provided in Appendix C.

## 4 RELATED WORK AND DISCUSSION

Bayesian methods capable of flexible nonlinear learning have been a topic of active study for the last several decades. Historically, early interest in Bayesian neural networks (MacKay, 1992; Neal, 1995) diminished as Gaussian processes rose to prominence (Rasmussen, 2004). In recent years, however, there has been growing interest in methods capable of learning expressive features, effectively quantifying uncertainty, and training efficiently on large datasets. Variational methods have seen particular attention in both neural networks (Blundell et al., 2015; Ovadia et al., 2019) and GPs (Hensman et al., 2013; Titsias, 2009; Liu et al., 2020) due to their flexibility and their ability to produce mini-batch gradient estimation training schemes.

While a wide range of work has aimed to produce more performant approximate Bayesian methods (including more expressive prior and posterior representations (Fortuin et al., 2021; Izmailov et al., 2021; Sun et al., 2019; Wilson & Izmailov, 2020)), they have still seen limited application, often due to the increased computational expense of these methods (Lakshminarayanan et al., 2017; Dusenberry et al., 2020). While some approaches to Bayesian neural networks have focused on improving the quality of the posterior uncertainty through e.g. better priors (Farquhar et al., 2020; Fortuin, 2022) or inference methods (Izmailov et al., 2021), other lines of work have focused on designing comparatively inexpensive approximate Bayesian methods. Indeed, simple strategies such as Bayesian dropout (Gal & Ghahramani, 2016) and stochastic weight averaging (Maddox et al., 2019) have seen much wider use than more expressive methods due to their simplicity.

One of the simplest Bayesian models is the BLL model that is the focus of this paper, which enables single-pass, often deterministic uncertainty prediction. This model has gained prominence through the lens of deep kernel learning (Wilson et al., 2016b;a; Watson et al., 2020; Liu et al., 2022) and within few-shot learning (Harrison et al., 2018; 2020; Harrison, 2021; Watson et al., 2021; Zhang et al., 2021). Deep kernel learning aims to augment standard neural network kernels with neural network inputs. This approach allows control of the behavior of uncertainty, particularly as a function of Euclidean distance (Liu et al., 2022). While stochastic variational inference has been applied to these models (Wilson et al., 2016a), efficient and deterministic mini-batch methods have not been a major focus. Moreover, classification in these models typically relies on sampling logits applying softmax functions, which increases variance (Ovadia et al., 2019; Kristiadi et al., 2020; 2021), or on Laplace approximation (Liu et al., 2022).

Within few-shot learning, exact conjugacy of the Bayesian linear regression model (Harrison et al., 2018) and Bayesian GDA (Harrison et al., 2020; Zhang et al., 2021; Snell et al., 2017) has been exploited for efficient few-shot adaptation. These models have (in addition to Van Amersfoort et al. (2020) among others) shown the strong performance of GDA-based/radial basis function networks, especially on problems such as out of distribution detection, which we further highlight in this work. However, training these models (as well as the DKL methods discussed previously) relies on direct computation of the marginal likelihood. In contrast to prior work on DKL and few-shot learning, our approach achieves efficient and deterministic training and prediction through our variational objectives and through similarly exploiting conjugacy, and thus the added complexity compared to standard neural network models is minimal.

## 5 EXPERIMENTS

We investigate the three VBLL models, with both MAP and variational feature learning, in regression and classification tasks. A full description of all metrics used throughout this section and baseline methods is available in the appendix. To illustrate VBLL models, we show predictions on simple datasets in Figure 1. The left figure shows a regression VBLL model with variational features trained on the function $f(x) = cx^3$, with training data shown in red. This figure shows the behavior on

---

[3]Complexity for the regression case is $\mathcal{O}(N_y^2 + N_\phi^2)$.

Table 1: Results for UCI regression tasks.

| | BOSTON | | CONCRETE | | ENERGY | |
|---|---|---|---|---|---|---|
| | NLL ($\downarrow$) | RMSE ($\downarrow$) | NLL ($\downarrow$) | RMSE ($\downarrow$) | NLL ($\downarrow$) | RMSE ($\downarrow$) |
| VBLL | $2.55 \pm 0.06$ | $\mathbf{2.92 \pm 0.12}$ | $3.22 \pm 0.07$ | $5.09 \pm 0.13$ | $1.37 \pm 0.08$ | $0.87 \pm 0.04$ |
| GBLL | $2.90 \pm 0.05$ | $4.19 \pm 0.17$ | $3.09 \pm 0.03$ | $5.01 \pm 0.18$ | $\mathbf{0.69 \pm 0.03}$ | $\mathbf{0.46 \pm 0.02}$ |
| LDGBLL | $2.60 \pm 0.04$ | $3.38 \pm 0.18$ | $\mathbf{2.97 \pm 0.03}$ | $\mathbf{4.80 \pm 0.18}$ | $4.80 \pm 0.18$ | $0.50 \pm 0.02$ |
| MAP | $2.60 \pm 0.07$ | $3.02 \pm 0.17$ | $3.04 \pm 0.04$ | $\mathbf{4.75 \pm 0.12}$ | $1.44 \pm 0.09$ | $0.53 \pm 0.01$ |
| RBF GP | $\mathbf{2.41 \pm 0.06}$ | $2.83 \pm 0.16$ | $3.08 \pm 0.02$ | $5.62 \pm 0.13$ | $\mathbf{0.66 \pm 0.04}$ | $0.47 \pm 0.01$ |
| Dropout | $\mathbf{2.36 \pm 0.04}$ | $\mathbf{2.78 \pm 0.16}$ | $\mathbf{2.90 \pm 0.02}$ | $\mathbf{4.45 \pm 0.11}$ | $1.33 \pm 0.00$ | $0.53 \pm 0.01$ |
| Ensemble | $2.48 \pm 0.09$ | $\mathbf{2.79 \pm 0.17}$ | $3.04 \pm 0.08$ | $4.55 \pm 0.12$ | $\mathbf{0.58 \pm 0.07}$ | $\mathbf{0.41 \pm 0.02}$ |
| SWAG | $2.64 \pm 0.16$ | $3.08 \pm 0.35$ | $3.19 \pm 0.05$ | $5.50 \pm 0.16$ | $1.23 \pm 0.08$ | $0.93 \pm 0.09$ |
| BBB | $\mathbf{2.39 \pm 0.04}$ | $\mathbf{2.74 \pm 0.16}$ | $2.97 \pm 0.03$ | $4.80 \pm 0.13$ | $\mathbf{0.63 \pm 0.05}$ | $0.43 \pm 0.01$ |
| VBLL BBB | $2.59 \pm 0.07$ | $3.13 \pm 0.19$ | $3.36 \pm 0.22$ | $5.16 \pm 0.16$ | $1.35 \pm 0.15$ | $0.062 \pm 0.03$ |

Table 2: Further results for UCI regression tasks.

| | POWER | | WINE | | YACHT | |
|---|---|---|---|---|---|---|
| | NLL ($\downarrow$) | RMSE ($\downarrow$) | NLL ($\downarrow$) | RMSE ($\downarrow$) | NLL ($\downarrow$) | RMSE ($\downarrow$) |
| VBLL | $\mathbf{2.73 \pm 0.01}$ | $\mathbf{3.68 \pm 0.03}$ | $1.02 \pm 0.03$ | $0.65 \pm 0.01$ | $1.29 \pm 0.17$ | $0.86 \pm 0.17$ |
| GBLL | $2.77 \pm 0.01$ | $3.85 \pm 0.03$ | $1.02 \pm 0.01$ | $0.64 \pm 0.01$ | $1.67 \pm 0.11$ | $1.09 \pm 0.09$ |
| LDGBLL | $2.77 \pm 0.01$ | $3.85 \pm 0.04$ | $1.02 \pm 0.01$ | $0.64 \pm 0.01$ | $1.13 \pm 0.06$ | $0.75 \pm 0.10$ |
| MAP | $2.77 \pm 0.01$ | $3.81 \pm 0.04$ | $0.96 \pm 0.01$ | $0.63 \pm 0.01$ | $5.14 \pm 1.62$ | $0.94 \pm 0.09$ |
| RBF GP | $2.76 \pm 0.01$ | $3.72 \pm 0.04$ | $\mathbf{0.45 \pm 0.01}$ | $\mathbf{0.56 \pm 0.05}$ | $\mathbf{0.17 \pm 0.03}$ | $\mathbf{0.40 \pm 0.03}$ |
| Dropout | $2.80 \pm 0.01$ | $3.90 \pm 0.04$ | $0.93 \pm 0.01$ | $0.61 \pm 0.01$ | $1.82 \pm 0.01$ | $1.21 \pm 0.13$ |
| Ensemble | $\mathbf{2.70 \pm 0.01}$ | $\mathbf{3.59 \pm 0.04}$ | $0.95 \pm 0.01$ | $0.63 \pm 0.01$ | $0.35 \pm 0.07$ | $0.83 \pm 0.08$ |
| SWAG | $2.77 \pm 0.02$ | $3.85 \pm 0.05$ | $0.96 \pm 0.03$ | $0.63 \pm 0.01$ | $1.11 \pm 0.05$ | $1.13 \pm 0.20$ |
| BBB | $2.77 \pm 0.01$ | $3.86 \pm 0.04$ | $0.95 \pm 0.01$ | $0.63 \pm 0.01$ | $1.43 \pm 0.17$ | $1.10 \pm 0.11$ |
| VBLL BBB | $2.74 \pm 0.01$ | $3.73 \pm 0.04$ | $0.94 \pm 0.03$ | $0.61 \pm 0.01$ | $2.96 \pm 0.59$ | $0.79 \pm 0.05$ |

so-called *gap* datasets—so named because of the interval between subsets of the data. The VBLL model shows desirable increasing uncertainty between the intervals (Foong et al., 2019). The right figure shows the generative classification model (G-VBLL) on the half-moon dataset. In particular, we visualize the feature density for each class. Importantly, the density has high Euclidean distance sensitivity, which has been advocated by Liu et al. (2022) as a desirable feature for robustness and out of distribution detection.

## 5.1 REGRESSION

We investigate the performance of the regression VBLL models on UCI regression datasets (Dua & Graff, 2017), which are standard benchmarks for Bayesian neural network regression (Moberg et al., 2019; Ober & Rasmussen, 2019; Daxberger et al., 2021b; Watson et al., 2021; Kristiadi et al., 2021). Results are shown in Tables 1, 2. We include baseline models run in Watson et al. (2021), and we replicate their experimental procedure and hyperparameters as closely as possible (details in the appendix).

Our experiments show strong results for VBLL models across datasets. Of particular interest is the performance relative to the GBLL model, which is trained directly on the exact marginal likelihood within the Bayesian last layer model. There are several contributing factors: the prior parameters were jointly optimized with the feature weights in the GBLL model, whereas prior terms were fixed in our VBLL model, resulting in a stronger regularization effect. Moreover, exact Bayesian inference can perform poorly under model misspecification (Grünwald & Van Ommen, 2017), whereas variational Bayes has comparatively favorable robustness properties and asymptotics (Giordano et al., 2018; Wang & Blei, 2019), although the Gaussian process (GP) model generally also has strong performance across datasets. Finally, directly targeting the marginal likelihood (computed exactly within conjugate models such as BLL models) has been shown to induce substantial overfitting (Ober et al., 2021; Thakur et al., 2020; Harrison, 2021), which the variational approach may avoid due to worse inferential efficiency.

## 5.2 IMAGE CLASSIFICATION

To evaluate performance of VBLL models in classification, we train the discriminative (D-VBLL) and generative (G-VBLL) classification models on the CIFAR-10 and CIFAR-100 image classification task. Following Liu et al. (2022), all experiments utilize a Wide ResNet-28-10 backbone architecture. We investigate full training methods (without a post-training step), indicated with the method name in the top third of Tables 3, 4; post-training methods, indicated by pre-training method + post-training method, in the middle third of the Tables; and feature uncertainty, in the bottom third.

We evaluate out of distribution (OOD) detection performance using the Street View House Numbers (SVHN) (Netzer et al., 2011) as a far-OOD dataset for both datasets, and CIFAR-100 for CIFAR-10 (and vice-versa) as near-OOD datasets. In-distribution data normalization is used in both cases. The DNN, BBB, D-VBLL and D-VBLL BBB models use maximum softmax probability (Hendrycks &

Table 3: Results for Wide ResNet-28-10 on CIFAR-10.

| Method | Accuracy (↑) | ECE (↓) | NLL (↓) | SVHN AUC (↑) | CIFAR-100 AUC (↑) |
|---|---|---|---|---|---|
| DNN | $95.8 \pm 0.19$ | $0.028 \pm 0.028$ | $0.183 \pm 0.007$ | $0.946 \pm 0.005$ | $0.893 \pm 0.001$ |
| SNGP | $95.7 \pm 0.14$ | $0.017 \pm 0.003$ | $0.149 \pm 0.005$ | $0.960 \pm 0.004$ | $\mathbf{0.902 \pm 0.003}$ |
| D-VBLL | $\mathbf{96.4 \pm 0.12}$ | $0.022 \pm 0.001$ | $0.160 \pm 0.001$ | $\mathbf{0.969 \pm 0.004}$ | $\mathbf{0.900 \pm 0.004}$ |
| G-VBLL | $\mathbf{96.3 \pm 0.06}$ | $0.021 \pm 0.001$ | $0.174 \pm 0.002$ | $0.925 \pm 0.015$ | $0.804 \pm 0.006$ |
| DNN + LL Laplace | $\mathbf{96.3 \pm 0.03}$ | $\mathbf{0.010 \pm 0.001}$ | $\mathbf{0.133 \pm 0.003}$ | $0.965 \pm 0.010$ | $0.898 \pm 0.001$ |
| DNN + D-VBLL | $\mathbf{96.4 \pm 0.01}$ | $0.024 \pm 0.000$ | $0.176 \pm 0.000$ | $0.943 \pm 0.002$ | $0.895 \pm 0.000$ |
| DNN + G-VBLL | $\mathbf{96.4 \pm 0.01}$ | $0.035 \pm 0.000$ | $0.533 \pm 0.003$ | $0.729 \pm 0.004$ | $0.661 \pm 0.004$ |
| G-VBLL + MAP | – | – | – | $0.950 \pm 0.006$ | $0.893 \pm 0.003$ |
| Dropout | $95.7 \pm 0.13$ | $0.013 \pm 0.002$ | $0.145 \pm 0.004$ | $0.934 \pm 0.004$ | $0.903 \pm 0.001$ |
| Ensemble | $\mathbf{96.4 \pm 0.09}$ | $0.011 \pm 0.092$ | $\mathbf{0.124 \pm 0.001}$ | $0.947 \pm 0.002$ | $\mathbf{0.914 \pm 0.000}$ |
| BBB | $96.0 \pm 0.08$ | $0.033 \pm 0.001$ | $0.333 \pm 0.014$ | $0.957 \pm 0.004$ | $0.844 \pm 0.013$ |
| D-VBLL BBB | $95.9 \pm 0.15$ | $0.058 \pm 0.019$ | $0.238 \pm 0.036$ | $0.832 \pm 0.026$ | $0.744 \pm 0.010$ |
| G-VBLL BBB | $95.9 \pm 0.16$ | $\mathbf{0.009 \pm 0.001}$ | $0.229 \pm 0.010$ | $0.917 \pm 0.005$ | $0.779 \pm 0.009$ |

Table 4: Results for Wide ResNet-28-10 on CIFAR-100.

| Method | Accuracy (↑) | ECE (↓) | NLL (↓) | SVHN AUC (↑) | CIFAR-10 AUC (↑) |
|---|---|---|---|---|---|
| DNN | $80.4 \pm 0.29$ | $0.107 \pm 0.004$ | $0.941 \pm 0.016$ | $0.799 \pm 0.020$ | $0.795 \pm 0.001$ |
| SNGP | $80.3 \pm 0.23$ | $\mathbf{0.030 \pm 0.004}$ | $\mathbf{0.761 \pm 0.007}$ | $\mathbf{0.846 \pm 0.019}$ | $0.798 \pm 0.001$ |
| D-VBLL | $\mathbf{80.7 \pm 0.03}$ | $0.040 \pm 0.002$ | $0.913 \pm 0.011$ | $\mathbf{0.849 \pm 0.006}$ | $0.791 \pm 0.003$ |
| G-VBLL | $80.4 \pm 0.10$ | $0.051 \pm 0.003$ | $0.945 \pm 0.009$ | $0.767 \pm 0.055$ | $0.752 \pm 0.015$ |
| DNN + LL Laplace | $80.4 \pm 0.29$ | $0.210 \pm 0.018$ | $1.048 \pm 0.014$ | $0.834 \pm 0.014$ | $\mathbf{0.811 \pm 0.002}$ |
| DNN + D-VBLL | $\mathbf{80.7 \pm 0.02}$ | $0.063 \pm 0.000$ | $0.831 \pm 0.005$ | $0.843 \pm 0.001$ | $0.804 \pm 0.001$ |
| DNN + G-VBLL | $80.6 \pm 0.02$ | $0.186 \pm 0.003$ | $3.026 \pm 0.155$ | $0.638 \pm 0.021$ | $0.652 \pm 0.025$ |
| G-VBLL + MAP | – | – | – | $0.793 \pm 0.032$ | $0.765 \pm 0.008$ |
| Dropout | $80.2 \pm 0.22$ | $0.031 \pm 0.002$ | $0.762 \pm 0.008$ | $0.800 \pm 0.014$ | $0.797 \pm 0.002$ |
| Ensemble | $\mathbf{82.5 \pm 0.19}$ | $0.041 \pm 0.002$ | $\mathbf{0.674 \pm 0.004}$ | $0.812 \pm 0.007$ | $\mathbf{0.814 \pm 0.001}$ |
| BBB | $79.6 \pm 0.04$ | $0.127 \pm 0.002$ | $1.611 \pm 0.006$ | $0.809 \pm 0.060$ | $0.777 \pm 0.008$ |
| D-VBLL BBB | $77.6 \pm 0.17$ | $0.041 \pm 0.003$ | $1.169 \pm 0.018$ | $0.785 \pm 0.022$ | $0.756 \pm 0.002$ |
| G-VBLL BBB | $78.1 \pm 0.18$ | $0.046 \pm 0.002$ | $1.156 \pm 0.008$ | $0.832 \pm 0.023$ | $0.742 \pm 0.004$ |

Gimpel, 2016) as an OOD measure. The G-VBLL and G-VBLL BBB models use a normalized feature density. Two methods for this exist: G-VBLL and G-VBLL BBB both use the learned variational posteriors to compute feature likelihoods. However, the performance of this is relatively weak, as there is no guarantee that learned feature likelihoods correspond effectively to true embedding densities. Thus, we also investigate an approach in which we estimate distributions for fixed features after training. This method estimates noise covariances for each class using the trained features, similar to the approach used in Liu et al. (2022). We refer to this model as G-VBLL-MAP, as the approach corresponds to MAP noise covariance estimation. These estimated covariances often result in overly-confident predictions, and so we do not advocate for label prediction under these fit covariances, and do not include results for them. Appendix B.6 discusses OOD methods, and further experimental details are in Appendix D.

Tables 3, 4 summarize the CIFAR-10 and CIFAR-100 results. D-VBLL and G-VBLL report strong accuracy performance and competitive metrics for both ECE and NLL. D-VBLL in particular demonstrates strong accuracy results, as well as competitive (with SNGP) NLL and OOD detection ability. Despite its comparative simplicity, it outperforms SNGP on accuracy and OOD on CIFAR-10 and accuracy on CIFAR-100. It matches SNGP on OOD for CIFAR-100, and is competitive (although slightly worse) on ECE and NLL. Overall, D-VBLL models stand out for their strong performance relative to their complexity. They also perform well as post-training models, whereas G-VBLL performs is substantially degraded.

While models with MAP feature estimation show strong performance versus baseline models, the performance of variational feature learning models (BBB) is more mixed. In regression tasks, these models are competitive, while in classification the performance is worse than deterministic models. In both settings, we use default KL term weighting (one over the dataset size). This contrasts with the tempered/cold posterior effect (Kapoor et al., 2022; Wenzel et al., 2020; Izmailov et al., 2021; Aitchison, 2020), in which it has been observed that alternative weightings of the likelihood and the KL may outperform this one. This is attributable (in part) to two factors: data augmentation and stochastic regularization. In regression there is no data augmentation and the model is trained for substantially longer than deterministic models; in classification we use standard augmentation and our training is more limited. Thus, it is possible that classification BBB models are over-regularized. We investigate this question in more detail in the appendix.

## 5.3 SENTIMENT CLASSIFICATION WITH LLM FEATURES

We evaluate VBLL models for language modelling tasks using the IMDB Sentiment Classification Dataset (Maas et al., 2011). The IMDB dataset is a binary text classification task consisting of 25,000 polarized movie reviews for training and another 25,000 for testing. A pre-trained OPT-175B (Zhang et al., 2022) model is used for text feature extraction. Sequence embeddings are obtained from OPT as the last token output from the the final network layer. We train both the generative (G-VBLL) and

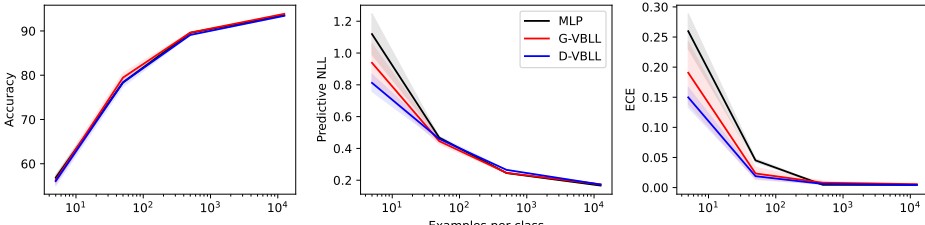

Figure 2: A performance comparison of G-VBLL, D-VBLL, and baseline MLP models on the IMDB Sentiment Classification Dataset. The models utilize text embeddings extracted from a pre-trained OPT-175B model. Results are presented across multiple training dataset scales, and the shaded regions represent $1\sigma$ error bounds.

Table 5: Wheel bandit cumulative regret.

|  | $\delta = 0.5$ | $\delta = 0.7$ | $\delta = 0.9$ | $\delta = 0.95$ | $\delta = 0.99$ |
|---|---|---|---|---|---|
| VBLL | **0.46 ± 0.01** | **0.89 ± 0.01** | **2.54 ± 0.02** | **4.82 ± 0.03** | **24.44 ± 0.71** |
| NeuralLinear | 1.10 ± 0.02 | 1.77 ± 0.03 | 4.32 ± 0.11 | 11.42 ± 0.97 | 52.64 ± 2.04 |
| NeuralLinear-MR | 0.95 ± 0.02 | 1.60 ± 0.03 | 4.65 ± 0.18 | 9.56 ± 0.36 | 49.63 ± 2.41 |
| LinDiagPost | 1.12 ± 0.03 | 1.80 ± 0.08 | 5.06 ± 0.14 | 8.99 ± 0.33 | 37.77 ± 2.18 |

Table 6: Wheel bandit simple regret.

|  | $\delta = 0.5$ | $\delta = 0.7$ | $\delta = 0.9$ | $\delta = 0.95$ | $\delta = 0.99$ |
|---|---|---|---|---|---|
| VBLL | **0.27 ± 0.03** | **0.69 ± 0.06** | 2.28 ± 0.14 | 4.16 ± 0.17 | **21.05 ± 1.59** |
| NeuralLinear | 0.31 ± 0.03 | **0.68 ± 0.07** | 2.18 ± 0.13 | 5.44 ± 0.73 | 46.42 ± 3.45 |
| NeuralLinear-MR | 0.33 ± 0.04 | 0.79 ± 0.07 | **2.17 ± 0.14** | **4.08 ± 0.20** | 35.89 ± 2.98 |
| LinPost-MR | 0.70 ± 0.06 | 0.99 ± 0.10 | 3.08 ± 0.22 | 4.85 ± 0.27 | 25.42 ± 1.81 |

discriminative (D-VBLL) models and a baseline MLP on the sequence embeddings via supervised learning at multiple training dataset scales: 10, 100, 1000 and 25,000 training samples. Evaluation is performed using the complete test set at each training dataset scale. Results are shown in Figure 2. The VBLL models demonstrate strong performance in comparison to the MLP baseline. We see significantly lower predictive NLL and ECE at smaller training dataset sizes. These findings validate the VBLL models' potential for integration with large-scale modern language models for diverse applications, particularly in sentiment classification tasks.

## 5.4 WHEEL BANDIT

To investigate the value of VBLL models in an active learning setting, we apply a VBLL regression model to the wheel bandit problem presented in Riquelme et al. (2018). This problem is a contextual bandit in which the state is sampled randomly in a two dimensional ball, and the learned model aims to identify the reward function. There are five regions in the ball and five actions: each region roughly corresponds to a correct action yielding a high reward, and incorrect action choice yields a low reward, although action 1 always yields an intermediate reward and no high-reward action exists for region 1. The parameter $\delta$ controls the volume of the high-reward regions, with larger $\delta$ corresponding to smaller high-reward regions. We report both cumulative regret—the difference in reward compared to an oracle, normalized to the performance of a random agent, aggregated over the full problem duration—and the simple regret, which captures only the last 500 timesteps and thus (roughly) measures the final quality of the learned model. We use a Thompson sampling policy (Russo et al., 2018; Thompson, 1933), and compare to the top models reported in (Riquelme et al., 2018). We find that our VBLL model strongly outperforms the top performing baselines in cumulative regret (Table 5) and slightly outperforms them in simple regret (Table 6), implying both the capacity of the model matches the best baselines while also exploring more effectively.

## 6 CONCLUSIONS AND FUTURE WORK

We have presented a simple, nearly computationally free Bayesian last layer architecture that can be applied to arbitrary network backbones. The practical realization of the VBLL model is a small number of extra parameters (corresponding to the variational posterior covariance) and a small number of regularization terms corresponding to terms arising in the marginalized predictive likelihood, prior terms used in MAP estimation, and KL divergences. Several important directions for future work exist. First, few-show adaptation that further exploits conjugacy of these models via e.g. recursive Bayesian least squares is possible. We have only leveraged basic ideas from variational inference in this work; there are many highly practical ideas within variational Kalman filtering which may enable efficient model adaptation, label noise robustness, inference within heavy-tailed noise, or improved time series filtering (Sykacek & Roberts, 2002; Sarkka & Nummenmaa, 2009; Ting et al., 2007).

ACKNOWLEDGMENTS

We acknowledge Apoorva Sharma, Jascha Sohl-Dickstein, Alex Alemi, and Allan Zhou for useful conversations over the course of this work. We also gratefully acknowledge Paul Brunzema, who identified a subtle bug in our initial results.

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

## A  THE MULTIVARIATE REGRESSION MODEL

In the multivariate regression case, we consider a model of the form

$$\boldsymbol{y} = W\boldsymbol{\phi} + \boldsymbol{\varepsilon} \tag{21}$$

and place a matrix normal (Tiao & Zellner, 1964; Geisser, 1965) prior on $W$, with $W \sim \mathcal{MN}(\bar{\boldsymbol{w}}, I, S)$. For a discussion of the matrix normal distribution, we refer the reader to (Box & Tiao, 2011).

Given the matrix normal prior and the above model, the posterior is also matrix normal. We thus fix a matrix normal variational posterior. In Appendix F.2, we obtain an ELBO of the form

$$\mathcal{L}(\boldsymbol{\theta}, \boldsymbol{\eta}, \Sigma) = \frac{1}{T} \sum_{t=1}^{T} \left( \log \mathcal{N}(\boldsymbol{y}_t \mid \bar{W}\boldsymbol{\phi}_t, \Sigma) - \frac{1}{2} \boldsymbol{\phi}_t^\top S \boldsymbol{\phi}_t \mathrm{tr}(\Sigma^{-1}) \right). \tag{22}$$

for $\boldsymbol{\eta} = \{\bar{W}, S\}$, and we use this as a training objective.

For a parameter distribution $\mathcal{MN}(\bar{W}, I, S)$, prediction in this model is analytically tractable and is

$$p(\boldsymbol{y}_t \mid \boldsymbol{x}_t, \boldsymbol{\eta}, \boldsymbol{\theta}) = \mathcal{N}(\bar{W}\boldsymbol{\phi}, \boldsymbol{\phi}_t^\top S \boldsymbol{\phi}_t I + \Sigma). \tag{23}$$

## B  ALGORITHMIC DETAILS

In this section we present concrete details on training VBLL models. We first describe the procedure for MAP estimation, last layer training on frozen features, and variational learning of features, as described in the paper body. We then discuss prior choice, describe the resultant regularization terms, and describe prediction and out of distribution detection within these models.

### B.1  FEATURE POINT ESTIMATION

We propose to train our models via joint variational inference for the last layer and MAP estimation of network weights (and noise covariance), yielding optimization problem

$$\boldsymbol{\theta}^*, \boldsymbol{\eta}^*, \Sigma^* = \arg\max_{\boldsymbol{\theta}, \boldsymbol{\eta}, \Sigma} \left\{ \mathcal{L}(\boldsymbol{\theta}, \boldsymbol{\eta}, \Sigma) + T^{-1}(\log p(\boldsymbol{\theta}) + \log p(\Sigma) - \mathrm{KL}(q(\boldsymbol{\xi} \mid \boldsymbol{\eta}) \| p(\boldsymbol{\xi}))) \right\}. \tag{24}$$

We will write the three terms on the RHS (scaled by $1/T$) as $\mathcal{R}(\boldsymbol{\theta}, \boldsymbol{\eta}, \Sigma)$. Reasonable priors for neural network weights have been discussed in several papers (Blundell et al., 2015; Pearce et al., 2020; Fortuin, 2022; Farquhar et al., 2020; Watson et al., 2020; Dusenberry et al., 2020; Nalisnick, 2018). In this work, we use simple isotropic Gaussian priors which yields a weight decay regularizer. While variational inference for the noise covariance is possible, we choose (MAP) point estimation to simplify the model. We use a standard inverse-Wishart prior; ignoring terms that vanish in gradient computation, we have likelihood

$$\log p(\Sigma) = \frac{\nu + N + 1}{2} \mathrm{logdet} \Sigma^{-1} - \frac{1}{2} \mathrm{tr}(M\Sigma^{-1}) \tag{25}$$

where $\Sigma$ is $N \times N$, $\nu > N - 1$ are the degrees of freedom and $M$ is the scale matrix. The terms $\nu, M$ are hyperparameters that are fixed.

### B.2  POST-TRAINING WITH VBLL LAYERS

In addition to jointly training the features and the last layer, we can train them independently. This is potentially desirable in several situations. For example, a model may already be trained, and it is useful to augment the model with uncertainty post-hoc. We propose to first train a model using a standard network architecture and a standard loss function, yielding $\boldsymbol{\theta}^*$ and $\bar{\boldsymbol{w}}^*$ (or $\bar{W}^*$ in the multivariate case). Given these quantities, the last layer is trained via

$$\boldsymbol{\eta}^*, \Sigma^* = \arg\max_{\boldsymbol{\eta}, \Sigma} \left\{ \mathcal{L}(\boldsymbol{\theta}^*, \boldsymbol{\eta}, \Sigma) + T^{-1}(\log p(\Sigma) - \mathrm{KL}(q(\boldsymbol{\xi} \mid \boldsymbol{\eta}) \| p(\boldsymbol{\xi}))) \right\}. \tag{26}$$

Practically, one can initialize the mean of the variational last layer (in the regression of discriminative classification case) with the last layer point estimate $\boldsymbol{w}^*$ from the first phase of training.

### B.3  COLLAPSED VARIATIONAL INFERENCE FOR BAYESIAN NEURAL NETWORKS

Stochastic variational approximations to the posterior over the network weights have previously been used for Bayesian learning (Blundell et al., 2015). In this section, we discussion computation of variational posterior $q(\boldsymbol{\theta})$, following the SVI methodology as discussed previously. Whereas our

---

**Algorithm 1** Variational BLL Training: Regression

---

**Require:** Training data $D = \{X, Y\}$, variational posterior initialization $\boldsymbol{\eta} = (\bar{\boldsymbol{w}}, S)$, number of train epochs $N$, minibatch size $B$, optimization algorithm opt$(\cdot)$.

1: **for** $i = 1$ to $N$ **do**
2:     Split dataset $D$ in to minibatches $D_j = (X_j, Y_j), \ j = 1, \ldots, \lfloor T/B \rfloor$.
3:     **for** $j = 1$ to $\lfloor T/B \rfloor$ **do**
4:         $\hat{\mathcal{L}}(\boldsymbol{\theta}, \boldsymbol{\eta}, \Sigma) \leftarrow \frac{1}{B} \sum_{(\boldsymbol{x}, \boldsymbol{y}) \in (X_j, Y_j)} (-\log p(\boldsymbol{y} \mid \boldsymbol{x}, \bar{\boldsymbol{w}}) + \frac{1}{2} \mathrm{tr}(\Sigma^{-1}) \boldsymbol{\phi}(\boldsymbol{x})^\top S \boldsymbol{\phi}(\boldsymbol{x}))$
5:         $\mathcal{R}(\boldsymbol{\theta}, \boldsymbol{\eta}, \Sigma) \leftarrow \frac{1}{T}(\mathrm{KL}(q(\boldsymbol{\xi} \mid \boldsymbol{\eta}) \| p(\boldsymbol{\xi} \mid \boldsymbol{\omega})) - \log p(\boldsymbol{\theta}) - \log p(\Sigma))$
6:         $\boldsymbol{\theta} \leftarrow \boldsymbol{\theta} - \mathrm{opt}(\nabla_{\boldsymbol{\theta}} \hat{\mathcal{L}}(\boldsymbol{\theta}, \boldsymbol{\eta}, \Sigma) + \nabla_{\boldsymbol{\theta}} \mathcal{R}(\boldsymbol{\theta}, \boldsymbol{\eta}, \Sigma))$
7:         $\boldsymbol{\eta} \leftarrow \boldsymbol{\eta} - \mathrm{opt}(\nabla_{\boldsymbol{\eta}} \hat{\mathcal{L}}(\boldsymbol{\theta}, \boldsymbol{\eta}, \Sigma) + \nabla_{\boldsymbol{\eta}} \mathcal{R}(\boldsymbol{\theta}, \boldsymbol{\eta}, \Sigma))$
8:         $\Sigma \leftarrow \Sigma - \mathrm{opt}(\nabla_{\Sigma} \hat{\mathcal{L}}(\boldsymbol{\theta}, \boldsymbol{\eta}, \Sigma) + \nabla_{\Sigma} \mathcal{R}(\boldsymbol{\theta}, \boldsymbol{\eta}, \Sigma))$
9:     **end for**
10: **end for**

---

approaches developed in the previous section were deterministic, SVI for all network weights is not possible via deterministic marginalization. Thus, computing

$$\mathbb{E}_{q(\boldsymbol{\theta})}[\log p(\boldsymbol{y} \mid \boldsymbol{x}, \boldsymbol{\theta})] \tag{27}$$

is typically approximated using Monte Carlo methods. In Blundell et al. (2015), the authors turn to the reparameterization gradient estimator (Kingma & Welling, 2014; Mohamed et al., 2020) to enable the computation of the (Monte Carlo estimator of the) gradient with respect to the parameters of the variational posterior. We could take a similar strategy for both $\boldsymbol{\xi}$ and $\boldsymbol{\theta}$, turning to sampling-based approximation. However, this sampling scheme yields both noisy gradient estimates and is expensive, as each sample corresponds to a full network evaluation. Our approach will instead marginalize the last layer and sample (some of) the other layers. This corresponds to Rao-Blackwellization (Rao, 1992; Blackwell, 1947) of the variational lower bound estimator, yielding lower variance gradient estimates.

We will choose a posterior that factorizes over the (last layer) parameters and weights, $q(\boldsymbol{\xi}, \boldsymbol{\theta} \mid \boldsymbol{\eta}) = q(\boldsymbol{\xi} \mid \boldsymbol{\eta_\xi}) q(\boldsymbol{\theta} \mid \boldsymbol{\eta_\theta})$. We also, in the discussion below, suppress dependence on $\Sigma$; in practice, we will turn to point estimation for this term. Note that further mean field factorizations for $q(\boldsymbol{\theta} \mid \boldsymbol{\eta})$ are typically employed. For example, Blundell et al. (2015) factorize the posterior over all weights in the neural network. Given this posterior approximation, we have

$$\log p(Y \mid X) \geq \mathbb{E}_{q(\boldsymbol{\theta} \mid \boldsymbol{\eta_\theta}) q(\boldsymbol{\xi} \mid \boldsymbol{\eta_\xi})}[\log p(Y \mid X, \boldsymbol{\xi}, \boldsymbol{\theta})] - \mathrm{KL}(q(\boldsymbol{\xi} \mid \boldsymbol{\eta_\xi}) \| p(\boldsymbol{\xi})) - \mathrm{KL}(q(\boldsymbol{\theta} \mid \boldsymbol{\eta_\theta}) \| p(\boldsymbol{\theta}))$$

$$\tag{28}$$

under the assumption that the prior $p(\boldsymbol{\xi}, \boldsymbol{\theta}) = p(\boldsymbol{\xi}) p(\boldsymbol{\theta})$ and thus

$$\frac{1}{T} \log p(Y \mid X) \geq \mathbb{E}_{q(\boldsymbol{\theta} \mid \boldsymbol{\eta_\theta})}[\mathcal{L}(\boldsymbol{\theta}, \boldsymbol{\eta_\xi})] - \frac{1}{T} \mathrm{KL}(q(\boldsymbol{\xi} \mid \boldsymbol{\eta_\xi}) \| p(\boldsymbol{\xi})) - \frac{1}{T} \mathrm{KL}(q(\boldsymbol{\theta} \mid \boldsymbol{\eta_\theta}) \| p(\boldsymbol{\theta})) \tag{29}$$

for the lower bounds $\mathcal{L}$ developed in Section 3. Thus, algorithmically, we first compute the inner expectation and then approximate the outer expectation with a sampling-based estimator.

## B.4 TRAINING

We now present our full training approach for the regression and classification settings. A detailed procedure for training the regression model with point features is shown in Algorithm 1. Generally, we will minimize the lower bounds we developed for each model. We note that $\mathcal{L}$ is a sum over data; following Blundell et al. (2015), we compute an (unbiased) estimator $\hat{\mathcal{L}}$ for this term with mini-batches.

The factorization of the ELBO over the data implies a mini-batch estimator for the gradient. Note that

$$\frac{1}{T} \sum_{t=1}^{T} \mathbb{E}_{q(\boldsymbol{\xi} \mid \boldsymbol{\eta})}[\log p(\boldsymbol{y}_t \mid \boldsymbol{x}_t, \boldsymbol{\xi}, \boldsymbol{\theta})] = \mathbb{E}_t \mathbb{E}_{q(\boldsymbol{\xi} \mid \boldsymbol{\eta})}[\log p(\boldsymbol{y}_t \mid \boldsymbol{x}_t, \boldsymbol{\xi}, \boldsymbol{\theta})] \tag{30}$$

where the outer expectation on the RHS is with respect to a uniform distribution over $t = 1, \ldots, T$. Note that this also holds for lower bound on the data likelihood, in the case of classification. We can construct a randomized estimator for this expectation based on sub-sampling the data, in our case in mini-batches. For a mini-batch of $B$ datapoints, this yields an estimator for the ELBO of the form

$$\hat{\mathcal{L}}(\boldsymbol{\theta}, \boldsymbol{\eta}, \Sigma) = \frac{1}{B} \sum_{t=1}^{B} \mathbb{E}_{q(\boldsymbol{\xi} \mid \boldsymbol{\eta})}[\log p(\boldsymbol{y}_t \mid \boldsymbol{x}_t, \boldsymbol{\xi}, \boldsymbol{\theta}, \Sigma)]. \tag{31}$$

In the classification case, this may be an inequality. Note that in the limit of infinite training data $(T \to \infty)$ the weight on the KL term goes to zero.

We have trained VBLL models with both momentum SGD and AdamW (Loshchilov & Hutter, 2017). While both work effectively, they result in different uncertainty representations far from the data. The interaction of VBLLs with the stochastic regularization associated with different optimizers is an important direction of future work. Practically, gradient clipping was necessary to stabilize late training, especially in the regression case. As the noise variance concentrates, gradient magnitude is highly sensitive to small perturbations to features, which can be rapidly destabilizing; gradient clipping was necessary and sufficient to prevent this destabilization. Beyond these details, training VBLL models did not differ from training normal models.

## B.5 PREDICTION AND MONITORING

For prediction with VBLL models, we predict directly using the variational posterior, exploiting the conjugate prediction results described in Section 2. For all three VBLL models, training objective computation and prediction can be reduced from cubic to quadratic complexity (in the last layer input width) by careful parameterization and computation. The assumptions required to achieve quadratic complexity for the first two models are minor. However, for the generative classification model, diagonal covariances must be assumed. We discuss complexity in the next section.

Training yields learned network weights $\boldsymbol{\theta}^*$ (or a variational posterior over these weights), noise covariance $\Sigma^*$, and last layer variational posterior parameters $\boldsymbol{\eta}^*$. To make predictions, there are two options. In the case of the regression and generative classification model, we may discard the variational posterior and leverage exact conjugacy. Under (Gaussian) distributional assumptions, exact posteriors may be computed with fixed features. However, exact last layer posteriors may be badly calibrated due to violation of distributional assumptions. Instead, we may make predictions under the variational posterior directly, under the assumption that $q(\boldsymbol{\xi} \mid \boldsymbol{\eta}^*) \approx p(\boldsymbol{\xi} \mid X, Y)$, yielding

$$p(\boldsymbol{y} \mid \boldsymbol{x}, X, Y) \approx \mathbb{E}_{q(\boldsymbol{\xi}|\boldsymbol{\eta}^*)}[p(\boldsymbol{y} \mid \boldsymbol{x}, \boldsymbol{\xi}, \boldsymbol{\theta}^*, \Sigma^*)] \tag{32}$$

where $(\boldsymbol{x}, \boldsymbol{y})$ denote a test point. For the discriminative classification model, only prediction under the variational posterior is possible, and in this model, sampling or an approximation (e.g. Laplace) may be used.

The generative classification case provides predicted class probability biases (the predicted probability of seeing a particular class before observing a label) through the Dirichlet posterior. In cases where a system designer believes there is likely to exist distributional shift between the training data and the evaluation conditions, predictions may be directly controlled by modifying this Dirichlet posterior.

For the variational feature approach, prediction can be done by sampling features and computing mixture distributions, yielding

$$p(\boldsymbol{y} \mid \boldsymbol{x}, X, Y) \approx \mathbb{E}_{q(\boldsymbol{\theta}|\boldsymbol{\eta}^*)}\mathbb{E}_{q(\boldsymbol{\xi}|\boldsymbol{\eta}^*)}[p(\boldsymbol{y} \mid \boldsymbol{x}, \boldsymbol{\xi}, \boldsymbol{\theta}, \Sigma^*)] \tag{33}$$

$$\approx \frac{1}{K}\sum_{k=1}^{K}\mathbb{E}_{q(\boldsymbol{\xi}|\boldsymbol{\eta}^*)}[p(\boldsymbol{y} \mid \boldsymbol{x}, \boldsymbol{\xi}, \boldsymbol{\theta}_k, \Sigma^*)] \tag{34}$$

for $\boldsymbol{\theta}_k$ sampled i.i.d. from the variational posterior. In the regression case, this averaging is straightforward. For the classification cases, we can average pre-softmax or post-softmax. For example, in the case of generative classification, both

$$p(\boldsymbol{y} \mid \boldsymbol{x}, X, Y) \approx \frac{1}{K}\sum_{k=1}^{K}\text{softmax}_{\boldsymbol{y}}(\log p(\boldsymbol{y} \mid X, Y) + \log \mathbb{E}_{q(\boldsymbol{\xi}|\boldsymbol{\eta}^*)}[p(\boldsymbol{x} \mid \boldsymbol{y}, \boldsymbol{\xi}, \boldsymbol{\theta}_k)]) \tag{35}$$

and

$$p(\boldsymbol{y} \mid \boldsymbol{x}, X, Y) \approx \text{softmax}_{\boldsymbol{y}}(\log p(\boldsymbol{y} \mid X, Y) + \log \frac{1}{K}\sum_{k=1}^{K}\mathbb{E}_{q(\boldsymbol{\xi}|\boldsymbol{\eta}^*)}[p(\boldsymbol{x} \mid \boldsymbol{y}, \boldsymbol{\xi}, \boldsymbol{\theta}_k)]) \tag{36}$$

are valid Monte Carlo estimators for the predictive density, and the same holds for the discriminative classifier. In practice, we typically use the former (in which we directly average the post-softmax samples) due to the relative implementation simplicity, although the latter is necessary for some forms of out of distribution detection. Note that in the latter estimator,

$$\log \frac{1}{K}\sum_{k}\boldsymbol{x}_k = \text{LSE}_k(\log \boldsymbol{x}_k) - \log K \tag{37}$$

for generic $\boldsymbol{x}_k$ and $\log K$ vanishes in the softmax and my therefore be ignored, and where the use of log-sum-exp improves numerical stability.

### B.6 OUT OF DISTRIBUTION DETECTION

A desirable feature of robust deep learning models is the ability to distinguish between in distribution and out of distribution (OOD) data. We use several metrics for OOD detection with VBLL models. For the discriminative VBLL, we follow Liu et al. (2022) and use the maximum softmax probability (Hendrycks & Gimpel, 2016) for an OOD measure. This is computed by sampling from the distribution over logits and passing these samples through the softmax, where they are averaged.

For the generative classification model, we can use the feature density

$$p(\boldsymbol{x} \mid X, Y) \approx \sum_{\boldsymbol{y}} \mathbb{E}_{\boldsymbol{\theta},\boldsymbol{\xi}}[p(\boldsymbol{x}, \boldsymbol{y} \mid \boldsymbol{\xi}, \boldsymbol{\theta})] \tag{38}$$

as an OOD measure. In the above, the expectation are with respect to the variational posteriors; for the MAP estimation case, this corresponds to direct evaluation.

In practice, we found post-training noise covariance calibration improved OOD detection performance for the G-VBLL model. More precisely, we aim to replace a shared diagonal $\Sigma$ across all classes with a $\Sigma_{\boldsymbol{y}}$ for each class. Our intuition is that while the $\Sigma$ that is used in training is *prescriptive*—in the sense that it provides a model within which learning occurs—the estimated per-class $\Sigma_{\boldsymbol{y}}$ are *descriptive* of the accuracy of modelling during training. Indeed, the training objective for the G-VBLL model is label (marginal) predictive likelihood, and so the training signal to model class feature densities highly accurately is weak.

Our calibration procedure is as follows. First, we assume a (MAP) point estimate for feature means $\boldsymbol{\mu}_{\boldsymbol{y}}$. For sufficiently large datasets $S_{\boldsymbol{y}}$ rapidly concentrates, so the impact of this assumption is relatively minor. For each class, we then compute the MAP noise covaraince $\Sigma_{\boldsymbol{y}}$ under the inverse-Wishart prior. Concretely, the mean under Gaussian prior $\mathcal{N}(\bar{\boldsymbol{\mu}}, \Sigma)$ and known noise covariance $\Sigma$ is

$$\boldsymbol{\mu}_{\boldsymbol{y}} = (\Sigma_{\boldsymbol{y}}^{-1} + T_y \Sigma^{-1})(\Sigma^{-1} \sum \boldsymbol{\phi}_t + \Sigma_{\boldsymbol{y}}^{-1} \bar{\boldsymbol{\mu}}_{\boldsymbol{y}}) \tag{39}$$

$$= (\frac{1}{T_y} \Sigma_{\boldsymbol{y}}^{-1} \Sigma + I)(\frac{1}{T_y} \sum \boldsymbol{\phi}_t + \Sigma \Sigma_{\boldsymbol{y}}^{-1} \bar{\boldsymbol{\mu}}_{\boldsymbol{y}}) \tag{40}$$

where $T_y$ is the number of class occurrences for class $\boldsymbol{y}$ and where the sum is over all inputs in class $\boldsymbol{y}$. For sufficiently large $T$ and zero mean prior, this mean is approximately equal to the empirical average $\frac{1}{T} \sum \boldsymbol{x}_t$. Thus, taking $\hat{\boldsymbol{\mu}} = T^{-1} \sum \boldsymbol{x}_t$, the noise covariance can be estimated as

$$\hat{\Sigma}_{\boldsymbol{y}} = \frac{1}{T_y + \nu + N + 1}(M + \sum (\boldsymbol{\phi}_t - \hat{\boldsymbol{\mu}}_{\boldsymbol{y}})(\boldsymbol{\phi}_t - \hat{\boldsymbol{\mu}}_{\boldsymbol{y}})^\top) \tag{41}$$

which corresponds to the MAP posterior with a known mean, and where the sum is again over all inputs in class $\boldsymbol{y}$.

We note that while our strategy of sequentially estimating two MAP estimates is relatively unsophisticated, it is straightforward and yields good results, and is consistent for large datasets (under straightforward distributional assumptions). In the above, $N$ corresponds to the dimension of the covariance matrix (as in (25)) and $\nu$ and $M$ corresponds to the prior degrees of freedom and scale matrix, respectively. We found that this MAP covariances estimation outperformed the max likelihood covariance estimation as performed in Liu et al. (2022). Moreover, we note that both the empirical mean of the features for each class and the covariance can be recursively estimated in one pass over the data, and so the complexity of this step is $\mathcal{O}(T)$. Inspired by Ren et al. (2019; 2021), we subtract the log density under the feature prior as a normalization strategy, which also slightly improves performance.

While this post-training last layer posterior improves OOD performance, it is substantially over-concentrated for label prediction, yielding to dramatically over-confident predictions. It is an open question how to best estimate the last layer posterior to achieve both effective and calibrated label and OOD prediction.

## C PARAMETERIZATION, COMPLEXITY, REGULARIZATION, AND HYPERPARAMETERS

In this section, we discuss how to parameterize each of the terms appearing in each type of VBLL. In each model, we use a "mixed" parameterization—in contrast to the standard parameterization or natural parameterization. More precisely, we will parameterize the inverse noise covariance $\Sigma^{-1}$ and the covariance of the variational posterior $S$ via Cholesky factorizations, and directly parameterize means $\bar{W}, \boldsymbol{\mu}$. In our (limited) comparisons of the performance of different parameterizations, we

found that our mixed parameterization performed equivalently (if slightly better) to the standard parameterization, and both performed better than natural parameterization. Interestingly, this stands in contrast to standard practice in variational Gaussian process learning (Hensman et al., 2013), in which authors frequently aim to derive natural gradient optimization algorithms.

We will show that for each VBLL model, under a set of reasonable assumptions, complexity is at worst quadratic in the last layer width and linear in the output dimension. These complexity results enable use of VBLL models on problems with high input dimensionality and high output dimensionality. Moreover, our mini-batch gradient estimation training objective results in (standard) linear complexity of gradient estimation in batch size, enabling training on much larger datasets than is possible with standard marginal likelihood objectives.

## C.1 Regression Complexity

Our analysis will focus on the multivariate case, for which the univariate outputs are a special case. We directly parameterize the mean $\bar{W} \in \mathbb{R}^{N_y \times N_\phi}$. The covariances are parameterized via Cholesky decomposition to guarantee positive semi-definiteness; in particular we parameterize

$$\Sigma^{-1} = LL^\top, \qquad\qquad L = L_d + \texttt{diag}(\exp(\boldsymbol{l})) \tag{42}$$

$$S = PP^\top, \qquad\qquad P = P_d + \texttt{diag}(\exp(\boldsymbol{p})). \tag{43}$$

Where $P, L$ are lower triangular with positive diagonals, and thus $L_d, P_d$ are lower triangular with zero diagonal, and vector $\boldsymbol{l}, \boldsymbol{p}$ control diagonal elements.

Given these parameterizations, we show the complexity of each operation required for training is at most quadratic in $N_\phi$. The training objective has two terms: the log Gaussian density and the trace term. For the log density, we have

$$\boldsymbol{e}^\top \Sigma^{-1} \boldsymbol{e} = \boldsymbol{e}^\top LL^\top \boldsymbol{e} \tag{44}$$

for $\boldsymbol{e} = \boldsymbol{y} - \bar{W}\boldsymbol{\phi}_t$. The term $L^\top \boldsymbol{e}$ can be computed in $\mathcal{O}(N_y^2)$ time. The second term is $\boldsymbol{\phi}_t^\top S \boldsymbol{\phi}_t \text{tr}(\Sigma^{-1})$, for which $\boldsymbol{\phi}_t^\top S \boldsymbol{\phi}_t$ can be computed in $\mathcal{O}(N_\phi^2)$ time, and the trace term

$$\text{tr}(\Sigma^{-1}) = \text{tr}(LL^\top) = \|L\|_F^2 \tag{45}$$

which can be computed in $\mathcal{O}(N_y^2)$ time via squaring and summing the elements of $L$.

The remaining terms are the KL penalty on the variational posterior, and the inverse-Wishart prior on the noise covariance. Fixing a prior $\mathcal{MN}(\underline{\bar{w}}, I, \underline{S})$, the KL penalty for the multivariate regression case is (ignoring constants)

$$KL(q(\boldsymbol{\xi} \mid \boldsymbol{\eta})\|p(\boldsymbol{\xi})) = \frac{1}{2}(\text{tr}((\bar{W} - \underline{\bar{W}})^\top (\bar{W} - \underline{\bar{W}})\underline{S}^{-1}) + N_y \text{tr}(\underline{S}^{-1}S) + N_y \log \frac{\det \underline{S}}{\det S}) \tag{46}$$

We will fix an isotropic prior, $\underline{S} = sI$ for $s > 0$. Thus, the first term is

$$\text{tr}((\bar{W} - \underline{\bar{W}})^\top (\bar{W} - \underline{\bar{W}})\underline{S}^{-1}) = \frac{1}{s}\|\bar{W} - \underline{\bar{W}}\|_F^2 \tag{47}$$

with complexity $\mathcal{O}(N_y N_\phi)$, and the second term is

$$N_y \text{tr}(\underline{S}^{-1}S) = \frac{N_y}{s}\text{tr}(S) \tag{48}$$

where the trace can again be computed as the squared Frobenius norm of the Cholesky factor of $P$, for complexity $\mathcal{O}(N_\phi^2)$. The last term is

$$N_y \log \frac{\det \underline{S}}{\det S} = N_y N_\phi \log s - N_y \text{logdet} S \tag{49}$$

where $\text{logdet} S = 2\text{logdet}(P)$ which is equal to the sum of the log diagonal elements, which can be computed in $\mathcal{O}(N_\phi)$.

Finally, we have the inverse-Wishart noise covariance prior, which has terms $\text{logdet}\Sigma^{-1}$ and $\text{tr}(M\Sigma^{-1})$ for scale matrix $M$. The log determinant term may be computed as previously, with complexity $\mathcal{O}(N_y)$. Choosing scale matrix $M = mI$, we have $\text{tr}(M\Sigma^{-1}) = m\text{tr}(\Sigma^{-1})$ which again is $\mathcal{O}(N_y^2)$. Summing all of this up, we have the total complexity of VBLL computations as $\mathcal{O}(N_y^2 + N_\phi^2)$, which is equivalent to the complexity of standard matrix multiplication; thus, there is effectively zero added computational expense from the VBLL model compared to a standard network. The reader may easily verify that complexity of prediction is no greater than the training complexity in the regression model.

Table 7: Time per batch on CIFAR-10 training.

| Model | Run time (s) | % above DNN |
|---|---|---|
| DNN | 0.321 | 0% |
| D-VBLL | 0.338 | 5.2% |
| G-VBLL | 0.364 | 13.4% |

## C.2 CLASSIFICATION COMPLEXITY

The complexity for the discriminative classification model follows from the regression model. We use the same parameterization, although we turn to a diagonal noise covariance $\Sigma$. The computation of the KL penalty is identical to the regression case. The only difference is that $\phi_t^\top S_{\boldsymbol{y}} \phi_t$ must be computed for all classes $\boldsymbol{y}$, yielding complexity $\mathcal{O}(N_\phi^2 N_y)$. This term dominates the complexity of this model; however, further factorization of the covariance is straightforward and can reduce the practical complexity. To predict in these models, sampling realizations of the last layer must be done to sample logits. This sampling is straightforward to do using the Cholesky factorization of the covariance, and has quadratic complexity.

For the generative classification model, we are limited by the $\Sigma + S_{\boldsymbol{y}}$ term in the log-sum-exp. As far as we are aware, there is no (practical) way to compute this term with quadratic complexity, or otherwise inexpensively compute this log density. Thus, in this paper we restrict $\Sigma$ and $S$ to diagonal matrices, which results in linear complexity in $N_\phi$ for all operations in loss computation. Thus, under this approximate posterior, the complexity of the full training loss computation is $\mathcal{O}(N_\phi N_y)$, which is equivalent to standard neural network models. This covariance structure is relatively restrictive, and improvements may results from sparse covariance structures.

Concretely, we compare the run time of one step of training across a baseline DNN, and both flavors of VBLL. We compare these models on CIFAR-10 training on a NVIDIA T4 GPU, with the wide ResNet encoder used in the rest of the classification experiments. The results are shown in Table 7. We note that our VBLL implementations are not carefully optimized, and so these slowdowns are an upper bound on the possible slowdown.

## C.3 COMPLEXITY OF COMPARABLE BASELINES

There are a set of baseline methods that are similar to VBLLs but often have different complexity. As discussed throughout the paper, training BLL models by exploiting exact conjugacy (or exactly computing the marginal likelihood) requires iterating over the full training set, yielding linear complexity in the size of the dataset. This almost always makes standard marginal likelihood training intractable. More directly comparable is SNGP (Liu et al., 2022), which also exploits exact conjugacy (or approximation thereof for classification) but only computes the last layer covariance once per epoch. This amortizes the cost of iteration over the full dataset. In practice, they use an exponential moving average estimate of the covariance, which removes the need to load the data multiple times per epoch. However, this covariance must still be computed and inverted, which has cubic complexity in the last layer dimension. Last layer Laplace (Daxberger et al., 2021a) methods, similarly, require a pass over the full dataset and must invert a dense covariance matrix, yielding cubic complexity. However, this is only done as a post-processing step for a trained model.

## C.4 HYPERPARAMETERS

VBLL models introduce a small number of hyperparameters over standard network training. First, standard hyperparameters may need to be modified for VBLL models. For example, we found longer training runs resulted in slightly improved calibration, but we believe further investigation of learning rate schedules is necessary. For MAP features estimation, we use standard weight decay regularization values.

The main novel hyperparameters introduced by the VBLL model are those associated with priors. In particular, the last layer mean prior (defined by a mean and variance; in the regression case, these are written $\bar{w}, S$) must be chosen. Practically, it is common to normalize outputs to have isotropic Gaussian distributions for regression, and thus we have found $\bar{w} = 0$ and $S = I$ yield a reasonable if diffuse prior. For the classification case, we found these values similarly induce reasonable epistemic uncertainty over the predictive categorical distribution.

The other novel hyperparameters are those associated with the noise covariance inverse-Wishart prior, the degrees of freedom $\nu$ and the scale matrix $M$. For all experiments, we fix the scale matrix as a scalar multiple of the identity matrix, $M = mI$. In our regression experiments we fix these to be $(1, 1)$, and find good resulting performance, but further investigation is possible. In the classification case—and in particular the generative classification case—these parameters control the degree of

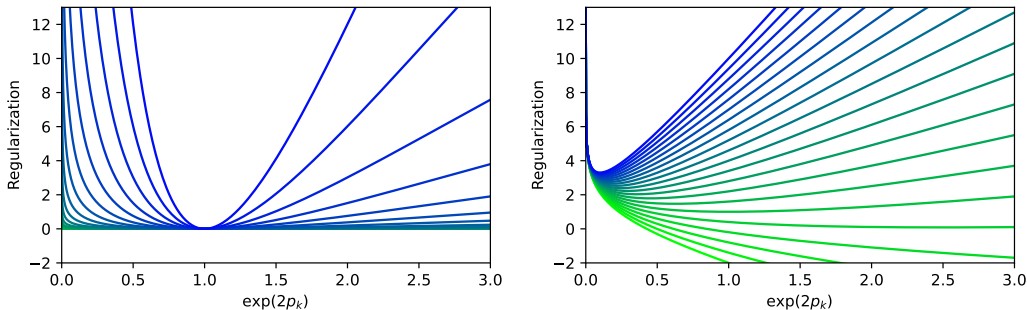

Figure 3: Weight decay (**left**) and our KL/Inverse-Wishart regularizers (**right**) plotted versus $\exp(\boldsymbol{p}_k)$ (which corresponds to the diagonal element of the covariance matrix). Different curves show varying weight decay strength and varying $a$ term in (56), with $b = 1$.

concentration in the feature space, and thus must be more carefully selected (and often co-selected with the weight decay strength).

### C.5 UNDERSTANDING PRIOR REGULARIZERS

In this subsection we investigate the regularization effects of the prior (and KL) terms, and compare them to standard regularizers such as weight decay. Note that naive weight decay on these parameterizations would correspond to additional loss terms of the form

$$\frac{\lambda}{2}(\|L_d\|_F^2 + \|\boldsymbol{l}\|_2^2 + \|P_d\|_F^2 + \|\boldsymbol{p}\|_2^2). \tag{50}$$

The loss terms resulting from our chosen priors in the regression case (and dropping terms with zero gradient) are

$$-\log p(\Sigma) = \frac{1}{2}(m\mathrm{tr}(\Sigma^{-1}) - \tilde{\nu}\mathrm{logdet}\Sigma^{-1}) \tag{51}$$

$$KL(q(\boldsymbol{\xi} \mid \boldsymbol{\eta})\|p(\boldsymbol{\xi})) = \frac{1}{2}\left(\frac{1}{s}\|\bar{W}\|_F^2 + \frac{N_y}{s}\mathrm{tr}(S) - N_y\mathrm{logdet}S\right) \tag{52}$$

for $\tilde{\nu} = \nu + N + 1$; note that (other than the weight decay-like term on $\bar{W}$) both covariance regularizers are of the form

$$a\mathrm{tr}(M) - b\mathrm{logdet}(M). \tag{53}$$

for constants $a, b$ and matrix $M$. Given our Cholesky parameterization,

$$\mathrm{tr}(\Sigma^{-1}) = \|P_d\|_F^2 + \sum_k \exp(2\boldsymbol{p}_k) \tag{54}$$

$$\mathrm{logdet}\Sigma^{-1} = \sum_k 2\boldsymbol{p}_k \tag{55}$$

and similarly for $S$. Thus, the regularization of the off-diagonal covariance terms again corresponds simply to weight decay, whereas the diagonal elements of both covariance matrices have regularizers of the form

$$\sum_k (a\exp(2\boldsymbol{p}_k) - 2b\boldsymbol{p}_k). \tag{56}$$

Note that this function is convex. This function (inside the summation) is visualized for varying $a$ (compared to weight decay) in Figure 3. Our regularization terms provide substantially more control over the minimizing value, and thus more control over predictive variance. However, compared to weight decay, our regularizers vary in scale substantially more which may lead to difficulties trading off regularization terms with other loss terms.

To counteract this relative lack of interpretability of our hyperparameters, we propose an alternate representation of these values. We rewrite the regularization function as

$$a\sum_k \left(\exp(2\boldsymbol{p}_k) - 2\frac{b}{a}\boldsymbol{p}_k\right). \tag{57}$$

where $a$ corresponds to a scale term, and $b/a$ controls the location of the minimum. We may specify a desired predictive variance, which can be mapped to the minimum of the regularization function. Concretely, given some target variance element $\hat{s} = \exp(2\boldsymbol{p}_k)$ (for all $k$), we choose

$$b = \hat{s}a \tag{58}$$

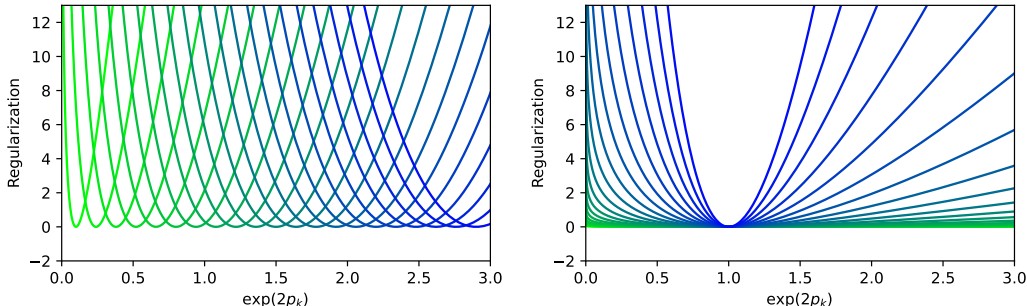

Figure 4: Sweeping over our modified hyperparameter representation. **Left**: sweeping over desired predictive variance $\hat{s}$, with $a = 100$. **Right**: sweeping over regularization scale $a$ with fixed desired predictive variance $\hat{s} = 1$. Note that all functions asymptote at $\exp(2\boldsymbol{p}_k) = 0$. In these figures, the curves have been vertically shifted to achieve a minimum at zero; this vertical shift does not impact regularization.

which assures that the minimum of (56) is achieved when $\boldsymbol{p}_k = \frac{1}{2}\log\hat{s}$ for all $k$. Sweeps over the hyperparameters $(a, \hat{s})$ are presented in Figure 4.

Given this transformation between hyperparameters, we can now be concrete in how to specify these alternate hyperparameters in VBLL models. The original hyperparameters for the model, as described earlier in this section, are the prior last layer covariance scale $s$, the scale matrix for the noise covariance prior $m$, the degrees of freedom $\tilde{\nu}$. Additionally, it is common is Bayesian deep learning to scale down the KL penalty, and we write this factor as $\lambda$. Our alternate hyperparameters are target (diagonal) values, $\hat{\boldsymbol{l}} > 0$ and $\hat{\boldsymbol{p}} > 0$, and scale parameters $\alpha_\Sigma > 0$ and $\alpha_S > 0$. The mapping between these hyperparameters is:

$$s \leftarrow \hat{\boldsymbol{p}} \qquad\qquad m \leftarrow \alpha_\Sigma \qquad\qquad (59)$$

$$\lambda \leftarrow \frac{\hat{\boldsymbol{p}}T\alpha_S}{N_y} \qquad\qquad \tilde{\nu} \leftarrow \hat{\boldsymbol{l}}\alpha_\Sigma. \qquad\qquad (60)$$

If $\lambda = 1$ as is (perhaps naively) theoretically justified in variational inference, then $\alpha_S$ is correspondingly fixed.

# D  EXPERIMENTAL DETAILS

This section contains details about the experiments in the body of the paper. We note that for highlighting in the tables in the paper body, if a single-pass method (in the upper half of each table) is the best performing in a metric, that result is highlighted. If the best performing is multi-pass, we highlight both the best multi-pass and single-pass method in the column. We believe that this is important, as many applications required single-pass methods and thus multi-pass results are irrelevant.

## D.1  METRICS

For regression experiments, we report the predictive negative log likelihood (NLL) of test data, which can be computed in closed form for point feature estimates. We also report the root mean squared error (RMSE), a standard metric for regression. For classification, in addition to the negative log likelihood, we also report predictive accuracy (based on standard argmax of the predictive distribution), and expected calibration error (ECE), which measures how the model's subjective predictive uncertainty agrees with predictive error. Finally, we also investigate out of distribution detection performance, a standard evaluation scheme for robust and probabilistic machine learning (Liu et al., 2022). We compute the area under the ROC curve (AUC) for near-OOD and far-OOD datasets, which is discussed in more detail later in this section.

## D.2  BASELINES

We distinguish baselines between single-pass and multi-pass models, which we show in upper and lower segments of each table, respectively. Single-pass methods require only a single network evaluation, and we compare VBLLs with MAP feature estimation to these models. Multi-pass methods require several network evaluations, and includes variational methods like Bayes-by-backprop (which we refer to as BBB) (Blundell et al., 2015), ensembles (Lakshminarayanan et al., 2017), Bayesian

dropout (Gal & Ghahramani, 2016) and stochastic weight averaging-Gaussian (SWAG) (Maddox et al., 2019).

Within regression, we compare to models which exploit exact conjugacy, including Bayesian last layer models (GBLL and LDGBLL (Watson et al., 2021)) and RBF kernel Gaussian processes. We note that these methods require computing full marginal likelihood and are thus difficult to scale to large training sets. We also compare to MAP learning, in which a full network is trained via MAP estimation, and a Bayesian last layer is fit to these fixed features (Snoek et al., 2015). Within classification, we primarily compare to standard networks (DNN), as these output a distribution over labels and thus can be directly compared to our model. We also compare to SNGP (Liu et al., 2022) and last layer Laplace-based methods (Daxberger et al., 2021a), which are similar last layer models. SNGP aims to approximate deep kernel GPs (Wilson et al., 2016b), and Laplace methods compute a last layer approximate posterior after training. We note that in contrast to SNGP (Liu et al., 2022), we do not modify a standard neural network backbone, such as including spectral normalization, adding residual connections, or using sinusoidal nonlinearities. Both SNGP and last layer Laplace methods require a pass over the full dataset to fit the last layer distribution; in contrast, our method maintains a last layer distribution during training, which may be useful for e.g. active learning. We do not evaluate Laplace methods in regression as they are nearly identical to the MAP model.

## D.3 TOY EXPERIMENTS

Figure 1 contains simple visualizations for the regression model and the generative VBLL model. In particular, the regression model shows predictions with variational feature learning (with KL weight of 1.0) on a cubic function with a gap in the data. This dataset consisted of 100 points sampled in $[-4, -2] \cup [2, 4]$, with a noise standard deviation of 0.1. The model consisted of a two hidden-layer MLP of width 128, trained for 1000 epochs with a batch size of 32, with stochastic gradient descent with momentum, with a learning rate of $3 \cdot 10^{-4}$, zero weight decay, and momentum beta parameters of 0.9. These values were arbitrarily chosen, although the choice of SGDM versus Adam (Kingma & Ba, 2015) does make a difference on prediction far from the data. Gradient clipping with a maximum magnitude of 2.0 was used. The DOF and scale parameters were both set to 1.0

For the classification problem, we used the scikit-learn (Pedregosa et al., 2011) implementation of the half moon dataset, with 1000 data points and a noise standard deviation of 0.2. We trained a G-VBLL model with residual-structured MLP of width 128 (each hidden layer is added to the layer input). This model was trained with SGDM with learning rate $3 \cdot 10^{-2}$, momentum beta 0.9, and weight decay $10^{-4}$, for 100 epochs and with a batch size of 32. The DOF parameter was 128, and the scale parameter was 1.0.

## D.4 REGRESSION

Our UCI experiments closely follow Watson et al. (2021), and we compare directly to their baselines. For VBLLs, we used a $\mathcal{N}(0, I)$ last layer mean prior and a $\mathcal{W}^{-1}(1, 1)$ noise covariance prior. For all experiments, we use the same MLP used in Watson et al. (2021) consisting of two layers of 50 hidden units each (not counting the last layer). For all datasets we matched Watson et al. (2021) and used a batch size of 32, other than the POWER dataset for which we used a batch size of 256 to accelerate training. For all datasets we normalize inputs (using the training set statistics) and subtract the training set means for the outputs. We did not re-scale the output magnitudes, to retain comparability of NLLs. We note that the extent to which outputs were normalized in Watson et al. (2021) is unclear. However, they make the parameters of their prior learnable, which can have a similar effect to centering the outputs, and so we believe our output centering is reasonable. All results shown in the body of the paper are for leaky ReLU activations. For all experiments, a fixed learning rate of 0.001 was used with the AdamW optimizer (Loshchilov & Hutter, 2017). A default weight decay of 0.01 was used for all experiments. We clipped gradients with a max magnitude of 1.0.

For all deterministic feature experiments, we ran 20 seeds. For each seed, we split the data in to train/val/test sets (0.72/0.18/0.1 of the data respectively). We train on the train set and monitor performance on the validation set to choose a total number of epochs. In contrast to Watson et al. (2021) who compute validation performance for every epoch, we compute validation performance (predictive NLL) every 10 epochs (note that the datasets are small and typically train for hundred of epochs). After choosing a number of epochs, we train on the combined training and validation set and evaluate performance on the test set. We use a max number of epochs shown in Table 8, which were large enough to not be reached but often lower than those used in Watson et al. (2021).

For our BBB feature models, we ran 10 seeds with a similar procedure to the above. We follow Watson et al. (2021) and use a $\mathcal{N}(0, 4/\sqrt{n_{in}})$ for each weight (where $n_{in}$ denotes the layer input

| Features | BOSTON | CONCRETE | ENERGY | POWER | WINE | YACHT |
|---|---|---|---|---|---|---|
| MAP | 3000 | 3000 | 2000 | 3000 | 1000 | 2000 |
| Variational | 10000 | 10000 | 10000 | 10000 | 10000 | 10000 |

Table 8: Maximum number of epochs for each set of features and each UCI dataset.

| | | Feature KL Weight | | |
|---|---|---|---|---|
| LL KL Weight | MAP | 50 | 5 | 0.5 |
| 1.0 | 0.160 | 0.266 | 0.281 | 0.282 |
| 0.1 | 0.162 | 0.266 | 0.286 | 0.272 |
| 0.01 | 0.168 | 0.268 | 0.268 | 0.280 |
| 0.001 | 0.160 | 0.267 | 0.272 | 0.276 |

Table 9: CIFAR-10 NLL for varying values of KL weights, for both the last layer and the feature weighting in variational feature learning.

width). Validation performance was monitored every 100 epochs, and 10 weight samples were used to compute the validation predictive likelihood and choose a full training number of epochs.

### D.5 IMAGE CLASSIFICATION

All classification experiments utilize the Wide ResNet-28-10 (WRN-28-10) backbone network architecture. Hyperparameters are similar to those proposed by Zagoruyko & Komodakis (2016). Unlike the original implementation of WRN, we do not employ Nesterov momentum and we fully decay an initial learning rate of 0.1 according to a Cosine Annealing schedule instead of a stepped decay schedule. Gradients are clipped with a maximum magnitude of 2.0 and we impose a last layer KL weight of 1.0. We All classification results are reported across 3 seeds and use the standard WRN data-augmentations proposed by (Zagoruyko & Komodakis, 2016). For the deterministic feature experiments, we train each model for 300 epochs.

The BBB backbone-based models utilize the same WRN architecture and are primarily deterministic. The BBB models implement a single final Bayesian linear layer with a prior distribution of $\mathcal{N}(0, 0.01)$. Each BBB-based model used 10 weight samples for test set evaluation. This operation is relatively cheap when compared to a fully stochastic network because the intermediate features are cached prior to the final Bayesian linear layer weight sampling and computation. All BBB are trained for 400 epochs and we impose a last layer KL weight of 1.0 and a feature KL weight of 0.5 the VBLL-BBB and DBLL-BBB models. The BBB baseline model utilized a feature KL weight of 50.

### D.6 SENTIMENT CLASSIFICATION WITH LLM FEATURES

We perform sentiment classification experiments utilizing features extracted from a pre-trained OPT-175B (Zhang et al., 2022) model on the IMDB Sentiment Classification dataset (Maas et al., 2011). We compare our G-VBLL and D-VBLL models with an MLP baseline. The IMDB dataset is a text-based binary classification task in which inputs are polarized movie reviews and outputs are positive and negative labels. Text embeddings are extracted from the OPT-175B model for each sample as the output of the last model layer for the final token in the text sequence. This results in a sequence embedding, $e = \mathbb{R}^{12288}$, for each sample. In all cases, we utilize two linear layers prior to the classification head. To understand the impact of training dataset size on performance, all experiments are performed at multiple training dataset scales. The IMDB dataset is sampled iid. to construct training datasets with 10, 100, 1000 samples alongside the standard 25,000 sample training split. We train models at all dataset scales and report across 3 seeds. The AdamW optimizer is used for all models. Hyperparameters such as learning rate, weight decay were tuned across both the 10 sample and full dataset scales.

### D.7 WHEEL BANDIT

We match the experimental settings of Riquelme et al. (2018). In particular, we use a batch size of 512, a learning rate of $3e - 3$, and train for 80000 steps total. We perform 20 steps in the environment per phase of updating, and perform 100 gradient steps when updating. We use a gradient clipping norm of $1.0$. We use the same network architecture as baselines, an MLP with widths $(100, 100, 5)$, where the last layer is a VBLL. The VBLL hyperparameters were set to defaults: the degrees of freedom and the scale in the Wishart prior are set to 1, and the prior scale was also set to 1.

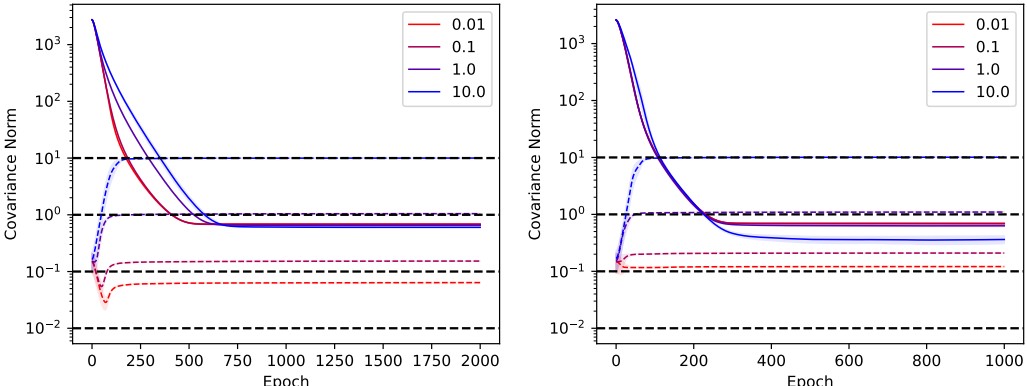

Figure 5: Sweeping over the $\Sigma$ location parameter for UCI datasets Energy (**left**) and Wine (**right**). The dotted colored lines correspond to $\Sigma^{-1}$ values over the course of training, and solid colored lines correspond to the Frobenius norm of $S$. The black dotted lines correspond to target $\Sigma^{-1}$ values. The scale hyparparameter was large in these experiments to illustrate the ability to effectively control noise covariance. Note that for very small $\Sigma^{-1}$, the impcat of the predictive loss limits the degree to which realized noise covariance matches the goal value; this trade-off is controlled by scale parameters.

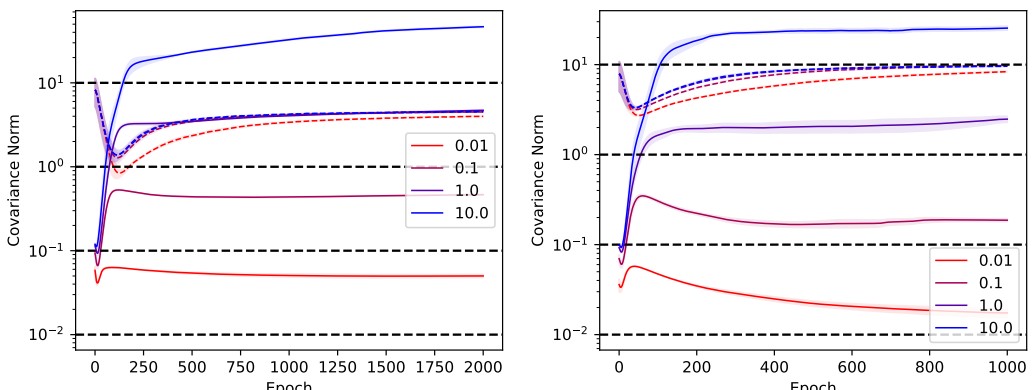

Figure 6: Sweeping over the $S$ location parameter for UCI datasets Energy (**left**) and Wine (**right**). Again, dotted colored lines correspond to $\Sigma^{-1}$ values over the course of training, solid colored lines correspond to the Frobenius norm of $S$, and black dotted lines correspond to target diagonal $S$ values. Note that the Frobenius norm of $S$ in all cases is higher than the target due to the off-diagonal elements, but the realized covariance can be well controlled.

## E  HYPERPARAMETER STUDIES AND ABLATIONS

**KL weight.**  We additionally explore the NLL sensitivity of the DBLL and DBLL-BBB models to various KL weighting configurations. In Table 9, we sweep across orders of magnitude for both the last layer and feature KL weighting parameters.

**Location and scale hyperparameters.**  We investigate our hyperparameter reformulation on UCI datasets in Figures 5 − 7. In particular, we vary each of the location and scale parameters and show that we can effectively control the quantities of interest. In particular, Figures 5 and 6 show varying the location hyperparameter for each covariance matrix $\Sigma, S$ with a high scale hyperparameter, enabling fine-grained control over realized values. In practice, this degree of direct control over realized model values is not desirable, and these plots only illustrate that such a degree of control is possible. In Figure 7, we vary the scale parameter for $\Sigma$ and show that it effectively controls the strength with which $\Sigma$ is regularized. With naive hyperparameter selection, interaction between scale and location parameters would require careful planning to control regularization scale independently of location, whereas our reformulation enables direct control of scale.

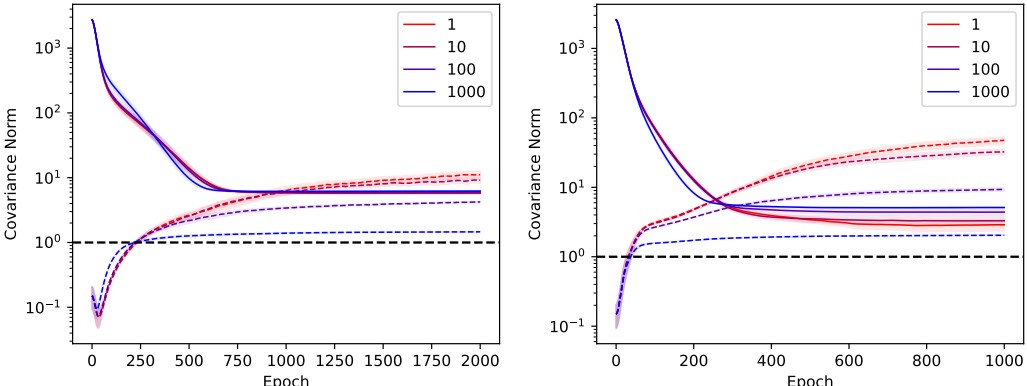

Figure 7: Sweeping over the $\Sigma$ scale parameter for UCI datasets Energy (**left**) and Wine (**right**). Again, dotted colored lines correspond to $\Sigma^{-1}$ values, solid colored lines correspond to the Frobenius norm of $S$, and the black dotted line corresponds to a location hyperparameter $\Sigma^{-1}$ value of 1. Note that by varying the scale hyperparameter, the strength of the regularization is varied without changing the target value, which is a result of our hyperparameter reformulation.

# F    PROOFS AND FURTHER THEORETICAL RESULTS

## F.1    HELPER RESULTS

Our first result builds on results from the variational Gaussian process literature (Titsias, 2009; Hensman et al., 2013).

**Lemma 4.** *Let* $q(\boldsymbol{\mu}) = \mathcal{N}(\bar{\boldsymbol{\mu}}, S)$ *and* $p(\boldsymbol{y} \mid X, \boldsymbol{\mu}) = \mathcal{N}(X\boldsymbol{\mu}, \Sigma)$ *with* $\boldsymbol{y} \in \mathbb{R}^N, \bar{\boldsymbol{\mu}}, \boldsymbol{\mu} \in \mathbb{R}^M,$ $X \in \mathbb{R}^{N \times M}$, *and* $S, \Sigma \in \mathbb{R}^{M \times M}$. *Then*

$$\mathbb{E}_{q(\boldsymbol{\mu})}[\log p(\boldsymbol{y} \mid X, \boldsymbol{\mu})] = \log p(\boldsymbol{y} \mid X, \bar{\boldsymbol{\mu}}) - \frac{1}{2}\mathrm{tr}(\Sigma^{-1}XSX^{\top}). \tag{61}$$

*Proof.* We have

$$\mathbb{E}_{q(\boldsymbol{\mu})}[\log p(\boldsymbol{y} \mid X\boldsymbol{\mu})] = -\frac{1}{2}\mathbb{E}_{q(\boldsymbol{\mu})}[\mathrm{logdet}(2\pi\Sigma) + (\boldsymbol{y} - X\boldsymbol{\mu})^{\top}\Sigma^{-1}(\boldsymbol{y} - X\boldsymbol{\mu})] \tag{62}$$

$$= -\frac{1}{2}\left(\mathrm{logdet}(2\pi\Sigma) + \mathbb{E}_{q(\boldsymbol{\mu})}[(\boldsymbol{y} - X\boldsymbol{\mu})^{\top}\Sigma^{-1}(\boldsymbol{y} - X\boldsymbol{\mu})]\right) \tag{63}$$

$$= -\frac{1}{2}\left(\mathrm{logdet}(2\pi\Sigma) + (\boldsymbol{y} - X\bar{\boldsymbol{\mu}})^{\top}\Sigma^{-1}(\boldsymbol{y} - X\bar{\boldsymbol{\mu}}) + \mathrm{tr}(\Sigma^{-1}XSX^{\top})\right) \tag{64}$$

where the last line follows from the fact that $\boldsymbol{y} - X\boldsymbol{\mu} \sim \mathcal{N}(\boldsymbol{y} - X\bar{\boldsymbol{\mu}}, XSX^{\top})$. The first two terms form the desired log density. $\square$

Based on this result, we can state a straightforward corollary for generative classification.

**Corollary 1.** *Let* $q(\boldsymbol{\mu}) = \mathcal{N}(\bar{\boldsymbol{\mu}}, S)$ *and* $p(\boldsymbol{y} \mid \boldsymbol{\mu}) = \mathcal{N}(\boldsymbol{\mu}, \Sigma)$ *with* $\boldsymbol{y}, \bar{\boldsymbol{\mu}}, \boldsymbol{\mu} \in \mathbb{R}^N$, $S, \Sigma \in \mathbb{R}^{N \times N}$. *Then*

$$\mathbb{E}_{q(\boldsymbol{\mu})}[\log p(\boldsymbol{y} \mid \boldsymbol{\mu})] = \log p(\boldsymbol{y} \mid \bar{\boldsymbol{\mu}}) - \frac{1}{2}\mathrm{tr}(\Sigma^{-1}S). \tag{65}$$

*Proof.* This result follows from Lemma 4 by simply choosing $X = I$. $\square$

We can also present a variant for multivariate classification.

**Corollary 2.** *Let* $q(W) = \mathcal{MN}(\bar{W}, I, S)$ *and* $p(\boldsymbol{y} \mid \boldsymbol{x}, W) = \mathcal{N}(W\boldsymbol{x}, \Sigma)$ *with* $\boldsymbol{y} \in \mathbb{R}^M, \bar{W}, W \in \mathbb{R}^{M \times N}$; $x \in \mathbb{R}^N$; $S \in \mathbb{R}^{N \times N}$; *and* $\Sigma \in \mathbb{R}^{M \times M}$. *Then*

$$\mathbb{E}_{q(W)}[\log p(\boldsymbol{y} \mid \boldsymbol{x}, W)] = \log p(\boldsymbol{y} \mid \boldsymbol{x}, \bar{W}) - \frac{1}{2}\boldsymbol{x}^{\top}S\boldsymbol{x}\,\mathrm{tr}(\Sigma^{-1}). \tag{66}$$

*Proof.* Our proof closely follows that of Lemma 4. Expanding the likelihood in the expectation, we have

$$\mathbb{E}_{q(W)}[\log p(\boldsymbol{y} \mid \boldsymbol{x}, W)] = \log p(\boldsymbol{y} \mid \boldsymbol{x}, \bar{W}) - \frac{1}{2}\mathbb{E}_W[\boldsymbol{x}^\top (W - \bar{W})^\top \Sigma^{-1}(W - \bar{W})\boldsymbol{x}] \quad (67)$$

Leveraging the matrix normal identity

$$\mathbb{E}_{W \sim \mathcal{MN}(\bar{W}, V, U)}[W^\top AW] = U\mathrm{tr}(A^\top V) + \bar{W}^\top A\bar{W} \quad (68)$$

and the fact that $W - \bar{W} \sim \mathcal{MN}(0, I, S)$, we have

$$\mathbb{E}[(W - \bar{W})^\top \Sigma^{-1}(W - \bar{W})] = S\mathrm{tr}(\Sigma^{-1}) \quad (69)$$

which completes the proof. $\square$

**Lemma 5.** *Let* $p(\boldsymbol{x} \mid \boldsymbol{\mu}) = \mathcal{N}(\boldsymbol{\mu}, \Sigma)$, *and let* $\boldsymbol{\mu} \sim \mathcal{N}(\bar{\boldsymbol{\mu}}, S)$. *Then,*

$$\mathbb{E}_{\boldsymbol{\mu}}[p(\boldsymbol{x} \mid \boldsymbol{\mu})] = \mathcal{N}(\bar{\boldsymbol{\mu}}, \Sigma + S). \quad (70)$$

*Proof.* We build upon Jacobson (1973) and note

$$\mathbb{E}_{\boldsymbol{x} \sim \mathcal{N}(\bar{\boldsymbol{\mu}}, S)}[\exp(-\frac{1}{2}\boldsymbol{x}^\top \Sigma^{-1}\boldsymbol{x})] = \sqrt{\frac{\det(S^{-1})}{\det(S^{-1} + \Sigma^{-1})}} \exp(-\frac{1}{2}\bar{\boldsymbol{\mu}}^\top S^{-1}(S - (\Sigma^{-1} + S^{-1})^{-1})S^{-1}\bar{\boldsymbol{\mu}}). \quad (71)$$

Note, by Woodbury's identity

$$S^{-1}(S - (\Sigma^{-1} + S^{-1})^{-1})S^{-1} = (S + \Sigma)^{-1} \quad (72)$$

Let $\boldsymbol{z} := \boldsymbol{x} - \boldsymbol{\mu}$, then $\boldsymbol{z} \sim \mathcal{N}(\boldsymbol{x} - \bar{\boldsymbol{\mu}}, S)$. We then have

$$\mathbb{E}[p(\boldsymbol{x} \mid \boldsymbol{\mu})] = \mathbb{E}[\exp(-\frac{1}{2}\|\boldsymbol{x} - \boldsymbol{\mu}\|_{\Sigma^{-1}}^2 + \frac{1}{2}\mathrm{logdet}(2\pi\Sigma^{-1}))] \quad (73)$$

$$= \mathbb{E}[\exp(-\frac{1}{2}\boldsymbol{z}^\top \Sigma^{-1}\boldsymbol{z})]]\exp(\frac{1}{2}\mathrm{logdet}(2\pi\Sigma^{-1})) \quad (74)$$

For the expectation we apply (71). We simplify the determinant term of (71) as

$$\sqrt{\frac{\det(S^{-1})}{\det(S^{-1} + \Sigma^{-1})}} = \exp(-\frac{1}{2}\mathrm{logdet}(I + S\Sigma^{-1})) \quad (75)$$

Combining, we have

$$\mathbb{E}[\exp(-\frac{1}{2}\boldsymbol{z}^\top \Sigma^{-1}\boldsymbol{z})]] = \exp(-\frac{1}{2}(\|\boldsymbol{x} - \bar{\boldsymbol{\mu}}\|_{(S+\Sigma)^{-1}}^2 + \mathrm{logdet}(I + S\Sigma^{-1})) \quad (76)$$

We have two log determinant terms, from (74) and the above. We can combine them as

$$\frac{1}{2}\mathrm{logdet}(2\pi\Sigma^{-1}) - \frac{1}{2}\mathrm{logdet}(I + S\Sigma^{-1}) = -\frac{1}{2}(\mathrm{logdet}(\frac{1}{2\pi}\Sigma) + \mathrm{logdet}(I + S\Sigma^{-1})) \quad (77)$$

$$= -\frac{1}{2}\mathrm{logdet}((\frac{1}{2\pi}\Sigma)(I + S\Sigma^{-1})) \quad (78)$$

$$= -\frac{1}{2}\mathrm{logdet}(\frac{1}{2\pi}\Sigma + \frac{1}{2\pi}S) \quad (79)$$

Combining all terms completes the proof. $\square$

## F.2 PROOF OF THEOREM 1

**Theorem 1.** *Let* $q(\boldsymbol{\xi} \mid \boldsymbol{\eta}) = \mathcal{N}(\bar{\boldsymbol{w}}, S)$ *denote the variational posterior for the BLL model defined in Section 2.1. Then,* (12) *holds with*

$$\mathcal{L}(\boldsymbol{\theta}, \boldsymbol{\eta}, \Sigma) = \frac{1}{T}\sum_{t=1}^{T}\left(\log \mathcal{N}(\boldsymbol{y}_t \mid \bar{\boldsymbol{w}}^\top \boldsymbol{\phi}_t, \Sigma) - \frac{1}{2}\boldsymbol{\phi}_t^\top S\boldsymbol{\phi}_t\Sigma^{-1}\right). \quad (80)$$

*Proof.* First,

$$\log p(Y \mid X, \boldsymbol{\theta}) = \log \mathbb{E}_{p(\boldsymbol{\xi})}[p(Y \mid X, \boldsymbol{\xi}, \boldsymbol{\theta})] \quad (81)$$

$$= \log \mathbb{E}_{q(\boldsymbol{\xi}|\boldsymbol{\eta})}[p(Y \mid X, \boldsymbol{\xi}, \boldsymbol{\theta})\frac{p(\boldsymbol{\xi})}{q(\boldsymbol{\xi} \mid \boldsymbol{\eta})}] \quad (82)$$

$$\geq \mathbb{E}_{q(\boldsymbol{\xi}|\boldsymbol{\eta})}[\log p(Y \mid X, \boldsymbol{\xi}, \boldsymbol{\theta})] - \mathrm{KL}(q(\boldsymbol{\xi} \mid \boldsymbol{\eta})\|p(\boldsymbol{\xi})) \quad (83)$$

$$= \sum_{t=1}^{T}\mathbb{E}_{q(\boldsymbol{\xi}|\boldsymbol{\eta})}[\log p(\boldsymbol{y}_t \mid \boldsymbol{x}_t, \boldsymbol{\xi}, \boldsymbol{\theta})] - \mathrm{KL}(q(\boldsymbol{\xi} \mid \boldsymbol{\eta})\|p(\boldsymbol{\xi})). \quad (84)$$

Note that the first term in the last line is the log of a Normal distribution. Applying Lemma 1, we have

$$\mathbb{E}_{q(\boldsymbol{\xi}|\boldsymbol{\eta})}[\log p(\boldsymbol{y}_t \mid \boldsymbol{x}_t, \boldsymbol{\xi}, \boldsymbol{\theta})] = \log p(\boldsymbol{y}_t \mid \boldsymbol{x}_t, \boldsymbol{\xi}, \boldsymbol{\theta}) - \frac{1}{2}\boldsymbol{\phi}_t^\top S \boldsymbol{\phi}_t \Sigma^{-1} \tag{85}$$

which completes the proof. □

We can also state the following corollary for the multivariate case.

**Corollary 3.** *Let $q(\boldsymbol{\xi} \mid \boldsymbol{\eta}) = \mathcal{MN}(\bar{W}, I, S)$ denote the variational posterior for the multivariate BLL model defined in Appendix A. Then, (12) holds with*

$$\mathcal{L}(\boldsymbol{\theta}, \boldsymbol{\eta}, \Sigma) = \frac{1}{T}\sum_{t=1}^{T}\left(\log \mathcal{N}(\boldsymbol{y}_t \mid \bar{W}\boldsymbol{\phi}_t, \Sigma) - \frac{1}{2}\boldsymbol{\phi}_t^\top S \boldsymbol{\phi}_t \mathrm{tr}(\Sigma^{-1})\right). \tag{86}$$

*Proof.* The proof follows the proof of Theorem 1, applying Corollary 2 instead of Lemma 1. □

### F.3 PROOF OF THEOREM 2

**Theorem 2.** *Let $q(W \mid \boldsymbol{\eta}) = \prod_{k=1}^{N_y}\mathcal{N}(\bar{\boldsymbol{w}}_k, S_k)$ denote the variational posterior for the discriminative classification model defined in Section 2.2. Then, (12) holds with*

$$\mathcal{L}(\boldsymbol{\theta}, \boldsymbol{\eta}, \Sigma) = \frac{1}{T}\sum_{t=1}^{T}\left(\boldsymbol{y}_t^\top \bar{W}\boldsymbol{\phi}_t - \mathrm{LSE}_k\left[\bar{\boldsymbol{w}}_k^\top \boldsymbol{\phi}_t + \frac{1}{2}(\boldsymbol{\phi}_t^\top S_k \boldsymbol{\phi}_t + \sigma_k^2)\right]\right) \tag{87}$$

*Proof.* We construct an ELBO via
$$\log p(Y \mid X, \boldsymbol{\theta}) = \log \mathbb{E}_{p(\boldsymbol{\xi})}[p(Y \mid X, \boldsymbol{\theta}, \boldsymbol{\xi})] \tag{88}$$

$$\geq \mathbb{E}_{q(\boldsymbol{\xi}|\boldsymbol{\eta})}[\log p(Y \mid X, \boldsymbol{\theta}, \boldsymbol{\xi})] - \mathrm{KL}(q(\boldsymbol{\xi} \mid \boldsymbol{\eta})||p(\boldsymbol{\xi})) \tag{89}$$

$$= \sum_{t=1}^{T}\mathbb{E}_{q(\boldsymbol{\xi}|\boldsymbol{\eta})}[\boldsymbol{y}_t^\top \log \mathrm{softmax}_{\boldsymbol{y}}(\log p(\boldsymbol{x}_t, \boldsymbol{y} \mid \boldsymbol{\theta}, \boldsymbol{\xi}))] - \mathrm{KL}(q(\boldsymbol{\xi} \mid \boldsymbol{\eta})||p(\boldsymbol{\xi})) \tag{90}$$

Expanding the log-softmax term, we have
$$\mathbb{E}_{q(\boldsymbol{\xi}|\boldsymbol{\eta})}\left[\boldsymbol{y}_t^\top \log \mathrm{softmax}_{\boldsymbol{y}}(\log p(\boldsymbol{x}_t, \boldsymbol{y} \mid \boldsymbol{\theta}, \boldsymbol{\xi}))\right] = \tag{91}$$
$$\mathbb{E}_{q(\boldsymbol{\xi}|\boldsymbol{\eta})}[\boldsymbol{y}_t^\top \log p(\boldsymbol{x}_t, \boldsymbol{y} \mid \boldsymbol{\theta}, \boldsymbol{\xi})] - \mathbb{E}_{q(\boldsymbol{\xi}|\boldsymbol{\eta})}[\mathrm{LSE}_{\boldsymbol{y}}[\log p(\boldsymbol{x}_t, \boldsymbol{y} \mid \boldsymbol{\theta}, \boldsymbol{\xi})].$$
As previously, under the variational posterior these likelihoods factorize across the data. The first term may be directly evaluated, yielding
$$\mathbb{E}_{q(\boldsymbol{\xi}|\boldsymbol{\eta})}[\log p(\boldsymbol{x}_t, \boldsymbol{y} \mid \boldsymbol{\theta}, \boldsymbol{\xi}))] = \mathbb{E}_{q(\boldsymbol{\xi}|\boldsymbol{\eta})}[\boldsymbol{w}_{\boldsymbol{y}}^\top]\boldsymbol{\phi} = \bar{\boldsymbol{w}}_{\boldsymbol{y}}^\top \boldsymbol{\phi}. \tag{92}$$
The second term (containing the log-sum-exp) can not be computed exactly, and so we will bound this term for both the discriminative and generative classifiers. Via Jensen's inequality, we have
$$-\mathbb{E}_{q(\boldsymbol{\xi}|\boldsymbol{\eta})}[\mathrm{LSE}_{\boldsymbol{y}}[\log p(\boldsymbol{x}_t, \boldsymbol{y} \mid \boldsymbol{\theta}, \boldsymbol{\xi})] \geq -\log \sum_{\boldsymbol{y}}\mathbb{E}_{q(\boldsymbol{\xi}|\boldsymbol{\eta})}[\exp(\log p(\boldsymbol{x}_t, \boldsymbol{y} \mid \boldsymbol{\theta}, \boldsymbol{\xi}))] \tag{93}$$

In the case of the discriminative model, we follow Blei & Lafferty (2007) and note that for each row $k$

$$\mathbb{E}_{\boldsymbol{w}_k \sim \mathcal{N}(\bar{\boldsymbol{w}}_k, S_k)}[\exp(\boldsymbol{w}_k^\top \boldsymbol{\phi}_t + \varepsilon_k)] = \exp(\bar{\boldsymbol{w}}_k^\top \boldsymbol{\phi}_t + \frac{1}{2}(\boldsymbol{\phi}_t^\top S_k \boldsymbol{\phi}_t + \sigma_k^2)) \tag{94}$$

which relies on assumed independence of rows of $W$ (although relaxation of this assumption is possible). Combining these results yields a lower bound on the ELBO, which is itself a lower bound on the marginal likelihood. □

### F.4 PROOF OF THEOREM 3

**Theorem 3.** *Let $q(\boldsymbol{\mu} \mid \boldsymbol{\eta}) = \prod_{k=1}^{N_y}\mathcal{N}(\bar{\boldsymbol{\mu}}_k, S_k)$ denote the variational posterior over class embeddings for the generative classification model defined in Section 2.3. Let $p(\boldsymbol{\rho} \mid Y) = Dir(\boldsymbol{\alpha})$ denote the exact Dirichlet posterior over class probabilities, with $\boldsymbol{\alpha}$ denoting the Dirichlet posterior concentration parameters. Then, (12) holds with*

$$\mathcal{L}(\boldsymbol{\theta}, \boldsymbol{\eta}, \Sigma) = \frac{1}{T}\sum_{t=1}^{T}(\log \mathcal{N}(\boldsymbol{\phi}_t \mid \bar{\boldsymbol{\mu}}_{\boldsymbol{y}_t}, \Sigma) - \frac{1}{2}\mathrm{tr}(\Sigma^{-1}S_{\boldsymbol{y}_t}) + \psi(\boldsymbol{\alpha}_{\boldsymbol{y}_t}) - \psi(\boldsymbol{\alpha}_*) + \log \boldsymbol{\alpha}_* \tag{95}$$

$$-\mathrm{LSE}_k[\log \mathcal{N}(\boldsymbol{\phi}_t \mid \bar{\boldsymbol{\mu}}_k, \Sigma + S_k) + \log \boldsymbol{\alpha}_k])$$
*where $\psi(\cdot)$ is the digamma function and where $\boldsymbol{\alpha}_* = \sum_k \boldsymbol{\alpha}_k$.*

*Proof.* Note that

$$\log p(Y \mid X, \boldsymbol{\theta}) \geq \mathbb{E}_{q(\boldsymbol{\xi}|\boldsymbol{\eta})}[\log p(Y \mid X, \boldsymbol{\xi}, \boldsymbol{\theta})] - \text{KL}(q(\boldsymbol{\xi} \mid \boldsymbol{\eta})||p(\boldsymbol{\xi})) \tag{96}$$

where

$$\mathbb{E}_{q(\boldsymbol{\xi}|\boldsymbol{\eta})}[\log p(Y \mid X, \boldsymbol{\xi}, \boldsymbol{\theta})] = \mathbb{E}_{q(\boldsymbol{\xi}|\boldsymbol{\eta})}\left[\log p(X \mid Y, \boldsymbol{\theta}, \boldsymbol{\xi}) - \log p(X \mid \boldsymbol{\theta}, \boldsymbol{\xi})\right] + \mathbb{E}_{q(\boldsymbol{\xi}|\boldsymbol{\eta})}[\log p(Y \mid \boldsymbol{\xi})] \tag{97}$$

All of these terms factorize over the data, as previously. We first note that for the last term,

$$\mathbb{E}_{\boldsymbol{\rho}}[\log p(\boldsymbol{y}_t \mid \boldsymbol{\theta}, \boldsymbol{\rho})] = \psi(\boldsymbol{\alpha}_{\boldsymbol{y}_t}) - \psi(\sum_{\boldsymbol{y}} \boldsymbol{\alpha}_{\boldsymbol{y}}) \tag{98}$$

where $\boldsymbol{\alpha}$ correspond to posterior Dirichlet concentration parameters and $\psi(\cdot)$ denotes the digamma function. The first term in (97) is the embedding likelihood; we can compute this expectation of the log likelihood via Corollary 1.

The second term in (97) is less straight-forward. Note that

$$\mathbb{E}[\log p(\boldsymbol{x}_t \mid \boldsymbol{\theta}, \boldsymbol{\xi})] = \mathbb{E}[\log \sum_{\boldsymbol{y}} p(\boldsymbol{x}_t \mid \boldsymbol{y}, \boldsymbol{\theta}, \boldsymbol{\xi})p(\boldsymbol{y} \mid \boldsymbol{\theta}, \boldsymbol{\xi})] \tag{99}$$

which can be written as a log-sum-exp of log joint likelihood. We will again apply Jensen's to exchange the log and sum, and note

$$-\mathbb{E}[\log p(\boldsymbol{x}_t \mid \boldsymbol{\theta}, \boldsymbol{\xi})] = -\mathbb{E}[\log \sum_{\boldsymbol{y}} p(\boldsymbol{x}_t \mid \boldsymbol{y}, \boldsymbol{\theta}, \boldsymbol{\xi})p(\boldsymbol{y} \mid \boldsymbol{\theta}, \boldsymbol{\xi})] \tag{100}$$

$$\geq -\log \mathbb{E}[\sum_{\boldsymbol{y}} p(\boldsymbol{x}_t \mid \boldsymbol{y}, \boldsymbol{\theta}, \boldsymbol{\xi})p(\boldsymbol{y} \mid \boldsymbol{\theta}, \boldsymbol{\xi})] \tag{101}$$

$$= -\log \sum_{\boldsymbol{y}} \mathbb{E}_{\boldsymbol{\mu}_{\boldsymbol{y}}}[p(\boldsymbol{x}_t \mid \boldsymbol{\mu}_{\boldsymbol{y}}, \boldsymbol{\theta})]\mathbb{E}_{\boldsymbol{\rho}}[p(\boldsymbol{y} \mid \boldsymbol{\rho})] \tag{102}$$

$$= -\text{LSE}_{\boldsymbol{y}}[\log \mathbb{E}_{\boldsymbol{\rho}}[p(\boldsymbol{y} \mid \boldsymbol{\rho})] + \log \mathbb{E}_{\boldsymbol{\mu}_{\boldsymbol{y}}}[p(\boldsymbol{x}_t \mid \boldsymbol{\mu}_{\boldsymbol{y}}, \boldsymbol{\theta})]] \tag{103}$$

where the second line follows from Jensen's, and the third line follows from the structure of the variational posterior. We may apply

$$\log \mathbb{E}_{\boldsymbol{\rho}}[p(\boldsymbol{y}_t \mid \boldsymbol{\theta}, \boldsymbol{\rho})] = \log \boldsymbol{\alpha}_{\boldsymbol{y}_t} - \log \sum_{\boldsymbol{y}} \boldsymbol{\alpha}_{\boldsymbol{y}}, \tag{104}$$

a standard result from Dirichlet-Categorical marginalization. The second term in (104) (the sum over concentration parameters) is equivalent for all classes $\boldsymbol{y}$, and thus can be pulled out of the log-sum-exp (due to the equivariance of this function under shifts) where it cancels the same third term in (97).

To compute the second expectation in (103), we apply Lemma 5. Combining all terms completes the proof. $\square$

