# OpenReview forum: "Variational Bayesian Last Layers"
_ICLR.cc/2024/Conference — ICLR 2024 spotlight_

### Official Review · Reviewer_b26R · 2023-10-23

**Soundness:** 4 excellent
**Presentation:** 4 excellent
**Contribution:** 4 excellent
**Rating:** 8
**Confidence:** 4

**Summary:**

This paper introduces Variational Bayesian Last layers, a technique to perform uncertainty estimation in standard neural network architectures.
The method performs bayesian learning only for the last layer in the neural networks using Variational Inference.
This results in a scalable and simple technique that shows strong performances in standard benchmarks for regression and classification.

**Strengths:**

1. I found the paper very interesting and easy to read. Uncertainty estimation is an important and active research area in deep learning.

1. To the best of my knowledge, the idea is novel (although not groundbreaking)

1. The method is scalable, simple to implement in standard architectures, and achieves very competitive performances (especially considering its simplicity)

1. While being Bayesian only on the last neural network layer the method is in principle not as powerful as other techniques, as the authors rightly claim simpler methods that are easy to implement are what is being more commonly used in practice (e.g. Bayesian dropout, stochastic weight averaging)

1. The appendix is extensive and addresses all the details I felt were missing in the main paper

**Weaknesses:**

I did not identify any major weaknesses, only some points for improvement

1. To increase the impact of the paper you need to make sure that people that are not too familiar with VI are able to easily implement the paper. This means:
    1. Make the code publicly available, especially to show how to best implement the "mixed parameterization" discussed in appendix D
    1. set good default hyperparameters

1. It would be useful to draw the graphical models of the models presented in Section 2, to help the reader visualize the random variables in play and their (hyper)priors/parameters

1. In case you need more space in the paper, I would move to the appendix some of the details on the generative classification model, especially considering the poorer performances.

**Questions:**

None, aside from the minor points presented in the weaknesses section

---

> ### Author Response · Authors · 2023-11-22
> **Response**
>
> We thank the reviewer for their very positive feedback.
>
> > To increase the impact of the paper you need to make sure that people that are not too familiar with VI are able to easily implement the paper. This means:
> > Make the code publicly available, especially to show how to best implement the "mixed parameterization" discussed in appendix D
> > set good default hyperparameters
>
> We are currently finalizing public releases of our code in both pytorch and jax, which will be released in time for the camera-ready version of the paper.
>
> > It would be useful to draw the graphical models of the models presented in Section 2, to help the reader visualize the random variables in play and their (hyper)priors/parameters
> > In case you need more space in the paper, I would move to the appendix some of the details on the generative classification model, especially considering the poorer performances.
>
> We thank the reviewer for these points. We have moved some details to the appendix, as noted.

---

### Official Review · Reviewer_tT7z · 2023-10-26

**Soundness:** 4 excellent
**Presentation:** 4 excellent
**Contribution:** 4 excellent
**Rating:** 10
**Confidence:** 4

**Summary:**

This paper derives efficient optimization objectives of the posterior distribution over the parameters of the last layer of neural networks for common machine learning applications such as classification, regression, and generative classification. Theoretically, their variational inference objective functions are derived in closed form and therefore enjoy the property of not requiring sampling - meaning the cost of enabling uncertainty quantification for a broad class of deep learning architectures is marginal (parameters other than the last layer are learned by maximum a posteriori). Experimentally, they validate these novel variational inference algorithms using standard benchmarks from UCI, a large language model used for sentiment analysis, and an image classification problem.

**Strengths:**

This is an excellent paper and a significant contribution - well done! The authors make clear how they build on the existing literature in Bayesian deep learning to create a novel advance that is practical and easy to implement. This is significant and should enable more work to push the frontier of the "best of both worlds", with neural networks serving as function approximators and Bayesian methods enabling sample-efficiency and quantification of uncertainty that is required for practical deployment of deep learning.

**Weaknesses:**

Visualizations of how tight or loose the bounds in the main text could help build more intuition; comparisons in terms of speed or efficiency to variational inference algorithms that do require sampling (such as Monte-Carlo objectives like VIMCO) could also help guide practitioners in making the correct trade-off depending on FLOPs of compute available versus the required accuracy of posterior approximation/uncertainty quantification.

**Questions:**

For the Resnet image recognition and sentiment analysis experiments, what was the additional compute required (or time taken per iteration, if available)? The sample-efficiency is great, and understanding the practical overhead rather than theoretical complexity would be great for larger models that are in broad use.

---

> ### Author Response · Authors · 2023-11-22
> **Response**
>
> We thank the reviewer for their very positive comments.
>
> > Visualizations of how tight or loose the bounds in the main text could help build more intuition; comparisons in terms of speed or efficiency to variational inference algorithms that do require sampling (such as Monte-Carlo objectives like VIMCO) could also help guide practitioners in making the correct trade-off depending on FLOPs of compute available versus the required accuracy of posterior approximation/uncertainty quantification.
>
> We conducted experiments to investigate tighter bounds on the softmax (based on Knowles and Minka, 2011), which did not meaningfully change performance. Thus, we believe that (surprisingly) the tightness of the bound does not meaningfully impact performance, and do not emphasize it. We believe complexity/run time versus performance is a more meaningful metric which we emphasize.
>
> > For the Resnet image recognition and sentiment analysis experiments, what was the additional compute required (or time taken per iteration, if available)? The sample-efficiency is great, and understanding the practical overhead rather than theoretical complexity would be great for larger models that are in broad use.
>
> We have included the runtime for our model compared to a standard model in for the ResNet.

---

### Official Review · Reviewer_vugz · 2023-10-27

**Soundness:** 2 fair
**Presentation:** 3 good
**Contribution:** 2 fair
**Rating:** 6
**Confidence:** 4

**Summary:**

The paper proposes a novel way to learn Bayesian last layer (BLL) networks by deriving a sampling-free approach for optimizing normal, categorical, and generative classification likelihoods,
by relying on variational inference techniques.
The approach is then evaluated on a series of regression and classification tasks against a range of baselines.


_____
_Edit: Given the improved presentation and evaluation, I increased my score._

**Strengths:**

BLL networks are an interesting approach to solve the scalability problem Bayesian neural networks tend to suffer from.
The paper introduces another variation to this family of approaches that is relatively straightforward, easy to understand, and implement.
The method is properly evaluated as the number of experimental setups is reasonably extensive both with respect to architectures and experimental tasks.

**Weaknesses:**

Straight-forward contributions can be seen both as a strength and as a weakness depending on the situation.
They are a strength if they are an easy solution to a complex problem that might not improve upon current approaches in all situations, but most.
They are a weakness if they do not provide a clear theoretical benefit above current approaches and also come without clear performance improvements.
For me, the results point to the latter case as they are rather mixed despite some strong wordings of the authors in their claims.

The abstract promises "improve[d] predictive accuracy, calibration, and out of distribution detection over baselines.", similarly in the contributions, and conclusion
parts of the paper. Even more, the method not only improves, but it also performs "extremely strong" and "exceptionally strong".
These are some exceptionally strong statements given the actual performance.

Focusing on each of the experiments in turn. The first problem is the presentation, e.g., what does a bold number mean (see question below)?

_Regression Experiments._ Of
the six data sets (see question below on this number) the proposal improves on two, slightly on one, equally on two (although better than neural net-based baselines), and worse than most of its baselines in the final one. Calling this "strong performance" is rather misleading. Two additional, though potentially minor, problems are that all of the baselines are simply cited from prior work (Watson et al., 2021). Given the wide performance variations between different train/test splits that can be observed for various UCI data sets the results are not entirely trustworthy. (Note that the reported error intervals are most likely standard errors, as is common on UCI, instead of standard deviations. But which they are is never specified.)
Secondly, the authors acknowledge in the appendix that there might be differences in the way the training data is normalized compared to the cited results.
(Whether these problems strongly influence the results, or bias them in favor or against VBLL is unclear.)

_Classification._
While "Extremely strong accuracy results" are mentioned, it just performs as well or worse than competitive baselines like Laplace or Ensembles. The same for ECE, NLL, OOD detection.  where "exceptionally strong performance" is claimed.

The method is somewhat simpler than baselines, but it lacks a convincing argument for why this should matter. As the authors advertise this simplicity, there should be additional results on practical runtime improvements compared to the baselines to provide some evidence for the claim that a reader should use this approach.

### Minor
- Dropout is first claimed to have a "substantially higher computational cost" (Sec 1) and appears later as a "comparatively inexpensive" method (Sec 4).  Additional forward passes at test time are indeed rather inexpensive instead of a high computational cost.
- When submitting to ICLR please make sure that you follow the ICLR style guide. E.g., Table captions belong above tables, not below.

### Typos
- Sec 5.2 first par: "We refer to this model **as** G-VBLL-MAP..."
- (6): $y|\rho$

**Questions:**

- Why was this specific subset of six UCI data sets chosen? The original work by Hernández-Lobato and Adams (2015), who introduced this set of experiments had ten, and even Watson et al. (2021) who the authors cite as relying on for their setup used ~~seven~~ different sets. _(PostRebuttal Edit: I misread the reference, Watson et al. use the full set of experiments.)_
- Can the authors provide further results on the empirical runtime of the proposed approach, not just a theoretical one?
- What was the principle according to which average numbers are bolded? E.g., in Energy RMSE a huge range of means is bold (from 0.39 to 0.47), but 0.43 is missing;  CIFAR-100 AUC has the same pattern, huge range, some missing, etc.
- (very minor) What is the irony in BLL methods being popular (Sec 2.4)?

---

> ### Author Response · Authors · 2023-11-22
> **Response**
>
> We thank the reviewer for their comments.
>
> > The abstract promises "improve[d] predictive accuracy, calibration, and out of distribution detection over baselines.", similarly in the contributions, and conclusion parts of the paper. Even more, the method not only improves, but it also performs "extremely strong" and "exceptionally strong". These are some exceptionally strong statements given the actual performance.
> Focusing on each of the experiments in turn. The first problem is the presentation, e.g., what does a bold number mean (see question below)?
>
> We understand that you feel the claimed contributions of the method are stated too strongly. We have revised our stated contributions to be more moderate.
>
> > Regression Experiments. Of the six data sets (see question below on this number) the proposal improves on two, slightly on one, equally on two (although better than neural net-based baselines), and worse than most of its baselines in the final one. Calling this "strong performance" is rather misleading. Two additional, though potentially minor, problems are that all of the baselines are simply cited from prior work (Watson et al., 2021). Given the wide performance variations between different train/test splits that can be observed for various UCI data sets the results are not entirely trustworthy. (Note that the reported error intervals are most likely standard errors, as is common on UCI, instead of standard deviations. But which they are is never specified.) Secondly, the authors acknowledge in the appendix that there might be differences in the way the training data is normalized compared to the cited results. (Whether these problems strongly influence the results, or bias them in favor or against VBLL is unclear.)
>
> We partially disagree with the reviewer about the strength of the method on regression experiments. The cases where our approach outperforms baselines, the outperformance is substantial. Otherwise, the method typically performs on par with the previous state of the art. Indeed, we believe our method is one of the largest jumps in performance on this task reported. Moreover, we note competitive baselines such as RBF GP and Ensembles are dramatically more expensive than our method.
>
> While we report baselines from (Watson et al. 2021), we note several things. First, part of the reason we built upon their results is the robustness of their experimental procedure. They perform 20 trials in which the train/val/test sets are chosen randomly, and the val set is used to automatically select the termination point of training. We follow this procedure in our experiments. On the topic of normalization, we believe our normalization matches that used in (Watson et al. 2021). In particular, they say that they use a zero-mean, unit-covariance prior “in whitened data space”. It is unclear if this denotes a direct transformation of their data (which would potentially skew reported NLLs) or indicates that prior terms are transformed to match the data. We center the data to be zero-mean, and normalize features to unit covariance in the input to the network, but do not re-scale the labels. We combine our re-scaling of the data with a zero-mean prior. Thus, comparing these strategies, we believe the practical effects of our data processing are nearly identical, although the effect of normalizing feature inputs to the network may vary.
>
> > Classification. While "Extremely strong accuracy results" are mentioned, it just performs as well or worse than competitive baselines like Laplace or Ensembles. The same for ECE, NLL, OOD detection. where "exceptionally strong performance" is claimed.
> The method is somewhat simpler than baselines, but it lacks a convincing argument for why this should matter. As the authors advertise this simplicity, there should be additional results on practical runtime improvements compared to the baselines to provide some evidence for the claim that a reader should use this approach.
>
> We have weakened our claims. While we do not strictly dominate in all metrics, the performance of our approach is highly competitive, which is especially notable given the simplicity of the approach.
>
> Direct comparison of runtimes for each model is difficult, because some methods are post-hoc only, some require a per-epoch step, and some require a change to each training step. Thus, timing depends on other training details such as batch size. We have included runtimes for our method in comparison to standard (non-Bayesian) network evaluation, and expanded our discussion of complexity for baseline methods.

---

> > ### Author Response · Authors · 2023-11-22
> > **Response (cont)**
> >
> > > Typos
> >
> > We have addressed these typos.
> >
> > > Why was this specific subset of six UCI data sets chosen? The original work by Hernández-Lobato and Adams (2015), who introduced this set of experiments had ten, and even Watson et al. (2021) who the authors cite as relying on for their setup used seven different sets.
> >
> > The choice was arbitrary, and they were simply chosen because they were easiest to get functional data loaders working for. Six datasets were chosen simply for spacing reasons.
> >
> > > Can the authors provide further results on the empirical runtime of the proposed approach, not just a theoretical one?
> >
> > As discussed above, we have included a comparison of the runtime of our method compared to a standard model.
> >
> > > What was the principle according to which average numbers are bolded? E.g., in Energy RMSE a huge range of means is bold (from 0.39 to 0.47), but 0.43 is missing; CIFAR-100 AUC has the same pattern, huge range, some missing, etc.
> >
> > The numbers were bolded as follows. First, results are split into two sections. The upper section corresponds to single-pass methods (in which only one network evaluation is required), and the lower section corresponds to multi-pass methods. For CIFAR experiments, we have also now included a middle section that consists of two-phsae, post-training methods. If the best method is single-pass, we highlight only this method. If the best method is a multi-pass method, we also highlight the best single-pass method. We believe this is fair, as the computational complexity of the two different classes of method are substantially different. Thus, if a practitioner is making a decision based on model cost and performance, the best single-pass method is still of interest.
> >
> > Two models are deemed to be tied if the second place mean is within the confidence interval associated with the best method. We note that one value in the UCI experiment tables was incorrectly not bolded, which is now fixed.
> >
> > > (very minor) What is the irony in BLL methods being popular (Sec 2.4)?
> >
> > Our goal in that statement was to highlight that while models have become increasingly large (and deep), shallow BLL models have demonstrated increasingly strong performance.

---

### Official Review · Reviewer_tM8T · 2023-10-30

**Soundness:** 3 good
**Presentation:** 3 good
**Contribution:** 3 good
**Rating:** 8
**Confidence:** 4

**Summary:**

The authors introduce variational Bayesian last layers as a novel approach for approximate inference in Bayesian deep learning models. The main contribution is three-fold: (i) following the current trend in Bayesian deep learning the authors propose to use a variational approximation to the last-layer posterior, (ii) the authors introduce closed-form bounds on the ELBO for different likelihood functions, (iii) the authors show that the simple approach can result in improved performance for regression and some classification tasks.

--

I have adjusted my score based on the author's response.

**Strengths:**

1. The paper is well-written and easy to follow in most parts. Moreover, the work is well-motivated and I enjoyed that the authors brought back old ideas to the BDL community, e.g., using the discriminant analysis as a likelihood model.
2. I believe the exposition of the method is well done in most places, though slightly dense here and there, and helped in understanding the general idea of the proposed method. Moreover, I believe that the method is correct and an interesting contribution to the field. I think it is important to see more work on deterministic approaches to uncertainty quantification in deep learning.
3. The experimental section shows promising results, especially in the case of regression.

**Weaknesses:**

Overall: My main concern with the paper is the weak empirical evaluation and limited novelty of the work, that is, it seems it is essentially an application of known techniques to the special case of last-layer posteriors.

Comments:
1. Section 2.4 lists various related works, which I believe the author claims to optimize the log marginal via gradient descent. I have not checked every citation, but it appears to me that this statement is false for at least a subset of the cited papers. It might be good to revise the exposition.
2. Eq 12 is some weighted ELBO, weighted with T for the purpose of generality, according to the authors. However, T never seems to be used later and makes the connection to the common ELBO less transparent. I believe the paper would improve in clarity if T is dropped.
3. Section 3.4 is very dense and it could help the reader if this section is improved in its presentation.
4. For the experiments, I would have expected assessments under distributional shift, a comparison to recent deterministic approaches (e.g., Zeng et al 2023 or Dhawan et al 2023), and a large-scale application of the approach as it acts on the last-layer only and should be applicable in more realistic scenarios (e.g., ImageNet).


Minor:
- Page 3, Eq 11 cites "Harrison et al 2018", which I looked up but didn't find any relevant content that would discuss the use of the marginal in Bayesian deep learning as a standard objective. What is the reason for the citation?

Citations:
- [Zeng et al 2023] Zeng et al, "Collapsed Inference for Bayesian Deep Learning", NeurIPS 2023.
- [Dhawan et al 2023] Dhawan et al, "Efficient Parametric Approximations of Neural Network Function Space Distance", ICML 2023.

**Questions:**

1. Eq. 14 uses a rather loose bound, is it possible that this is the reason the approach underperforms in the classification settings compared to the regression setting? If so, is there any way to obtain a tighter bound?
2. How does the method perform if it is used only as a post-hoc approach, meaning, without adaptation of the feature map? In large-scale applications, this is a particularly relevant setting and the proposed method could be a promising plug-in replacement.
3. From what I understand T is never actually used. Is this correct?

---

> ### Author Response · Authors · 2023-11-22
> **Response**
>
> We thank the reviewer for their comments, which we address point-by-point.
>
> > Overall: My main concern with the paper is the weak empirical evaluation and limited novelty of the work, that is, it seems it is essentially an application of known techniques to the special case of last-layer posteriors.
>
> We agree with the review that the methodological novelty may appear limited compared to other papers on Bayesian deep learning. However, our fundamental motivation is that the recent history of machine learning has made clear the importance of building inexpensive but effective methods, and the importance of nailing the details in methods. We believe that the depth of our work–derivation of novel deterministic training objectives, experimental analysis, treatment of complexity and parameterization, and analysis of hyperparameters–is critical to building useful systems that may be deployed at scale.
>
> We have added several elements to the experimental evaluation. In particular, we have added an evaluation of post-training methods (in which the VBLL is added to a model with frozen features) and added a bandit problem, in which VBLLs show extremely strong performance.
>
> > Section 2.4 lists various related works, which I believe the author claims to optimize the log marginal via gradient descent. I have not checked every citation, but it appears to me that this statement is false for at least a subset of the cited papers. It might be good to revise the exposition.
>
> We emphasize that the list of papers in section 2.4 are papers in which Bayesian last layer models have been used. Not all of these papers used the marginal likelihood objective for training these models; we have clarified this in the text.
>
> > Eq 12 is some weighted ELBO, weighted with T for the purpose of generality, according to the authors. However, T never seems to be used later and makes the connection to the common ELBO less transparent. I believe the paper would improve in clarity if T is dropped.
>
> The goal of including the factor of 1/T is to yield an objective that is more amenable to standard minibatch optimization, as is used in standard supervised learning. This is especially important to highlight when aggregation across minibatches should be done by averaging (as opposed to summation) and to weight the regularization terms. We have highlighted the importance of this.
>
> > Section 3.4 is very dense and it could help the reader if this section is improved in its presentation.
>
> We have made a collection of changes to the writing in this section to improve presentation, including splitting this section into two sections: one on training objectives and one on prediction and OOD.
>
> > For the experiments, I would have expected assessments under distributional shift, a comparison to recent deterministic approaches (e.g., Zeng et al 2023 or Dhawan et al 2023), and a large-scale application of the approach as it acts on the last-layer only and should be applicable in more realistic scenarios (e.g., ImageNet).
>
> We emphasize that, in terms of model parameter count, the wide ResNets are relatively large (~40M) parameters. Indeed, they are much larger compared to non-wide ResNets. We prioritized these experiments to enable comparison to established baselines such as SNGP. To include experiments on scalability, we also included the experiments in which we trained models on LLM features.
>
> To include experiments that are more similar to distribution shift (and also include elements of active learning to investigate the usefulness of the uncertainty quantification) we included bandit experiments, as described in the response to all reviewers.
>
> > Page 3, Eq 11 cites "Harrison et al 2018", which I looked up but didn't find any relevant content that would discuss the use of the marginal in Bayesian deep learning as a standard objective. What is the reason for the citation?
>
> [Harrison ea 2018] trains on the marginal likelihood, computed analytically in the regression case by iterating over the data, in the multi-task/few-shot learning setting. It also uses a Bayesian last layer approach that is structurally the same as our regression model.

---

> > ### Author Response · Authors · 2023-11-22
> > **Response (cont)**
> >
> > > Eq. 14 uses a rather loose bound, is it possible that this is the reason the approach underperforms in the classification settings compared to the regression setting? If so, is there any way to obtain a tighter bound?
> >
> > Several tighter bounds have been proposed, which we mostly did not address in the paper for simplicity. These are best discussed in:
> >
> > Non-conjugate Variational Message Passing for Multinomial and Binary Regression, Knowles and Minka, NeurIPS 2011.
> >
> > We investigated the adaptive bound from this paper, in which the bound is tightened by exploiting the shift equivariance of the LSE function. This bound introduces new variables which control the shifting of the bound, and can be trained along with other network weights.
> >
> > We found that the adaptive bound did not meaningfully change the performance of our method. Because the features are learned, it is likely that the automatic bound tightening enabled by Knowles and Minka’s bound was already happening with the simpler Jensen-only bound.
> >
> > > How does the method perform if it is used only as a post-hoc approach, meaning, without adaptation of the feature map? In large-scale applications, this is a particularly relevant setting and the proposed method could be a promising plug-in replacement.
> >
> > This is a very interesting question. We have included experiments in which we train a standard network, and then include a post-processing step in which we train the variational last layer only. This is highly relevant to many applications, including for models that have distinct pre-training and post-training steps.
> >
> > Broadly, we find that D-VBLL models perform well in this setting, while G-VBLL models perform worse. We believe the cause of this is that the data distribution in feature space matches between standard DNN training and D-VBLL models, whereas there is a mismatch between G-VBLL models and standard DNN training.
> >
> > We have included a substantial revision to the paper in which we discuss the post-training procedure.
> >
> > > From what I understand T is never actually used. Is this correct?
> >
> > T is an important term in the weighting of the regularization terms, and is an important feature of the algorithm. In particular, our experiments set the weighting of the KL term to be 1/T. We have clarified this in the paper.

---

> > > ### Comment · Reviewer_tM8T · 2023-11-23
> > >
> > > I thank the authors for the detailed response and for answering my questions/concerns. Consequently, I have changed my score to reflect the rebuttal and advocate for accepting the work.

---

### Author Response · Authors · 2023-11-22
**Response to all reviewers**

We thank all reviewers for their thoughtful reviews. We address each review as a response to that reviewer, but the main elements of our rebuttal are:
- We have conducted a set of new experiments. In particular:
  - To better investigate distribution shift and active learning, we have included a new set of results on the wheel bandit problem from the Bayesian bandit showdown paper (Riquelme et al., 2018). We find that VBLLs substantially outperform baselines on this problem, and we believe this is the most compelling demonstration so far of the importance of VBLLs.
  - We trained a VBLL on frozen features, in which we show the strong performance of (in particular) D-VBLL fine tuning on DNN features.
  - We investigated a tighter deterministic bound of the softmax term, and found that the tighter adaptive bound of Knowles and Minka (2011) does not result in performance changes.
  - We report runtimes for our model in comparison to a standard model; we find that VBLL models are 5% (G-VBLL) and 13% (D-VBLL) slower per training step than a standard DNN. We note that our implementation is not optimized, and so these numbers represent a cap on the slowdown from including VBLLs.
- We also include several writing changes, including a more complete discussion of computational complexity.
- We have answered the questions asked by each reviewer.

---

### Meta-Review · Area_Chair_rk9R · 2023-12-05

**Metareview:**

This paper introduces variational Bayesian last layers as an approach for approximate inference in Bayesian deep learning models. All four reviewers value the contributions and presentation of the method, and recommend the paper to be accepted. This is both a good and welcome addition to the conference program.

**Justification For Why Not Higher Score:**

I could see this paper as an oral as well, but I'm not sure how selective I should be on that.

**Justification For Why Not Lower Score:**

Given the enthusiasm the reviewers showed (and also my personal opinion), I would be happier to see this as a spotlight than just a poster.

---

### Decision · Program_Chairs · 2024-01-16

Accept (spotlight)